# Real-Time Selection Under General Constraints via Predictive Inference

**Yuyang Huo**[1*]    **Lin Lu**[1*]    **Haojie Ren**[2†]    **Changliang Zou**[1†]

[1]School of Statistics and Data Sciences, LPMC, KLMDASR and LEBPS,
Nankai University, Tianjin, China
[2]School of Mathematical Sciences, Shanghai Jiao Tong University, Shanghai, China
`huoyynk@gmail.com, linlu102099@gmail.com`
`haojieren@sjtu.edu.cn, zoucl@nankai.edu.cn`

## Abstract

Real-time decision-making gets more attention in the big data era. Here, we consider the problem of sample selection in the online setting, where one encounters a possibly infinite sequence of individuals collected over time with covariate information available. The goal is to select samples of interest that are characterized by their unobserved responses until the user-specified stopping time. We derive a new decision rule that enables us to find more preferable samples that meet practical requirements by simultaneously controlling two types of general constraints: individual and interactive constraints, which include the widely utilized False Selection Rate (FSR), cost limitations, diversity of selected samples, etc. The key elements of our approach involve quantifying the uncertainty of response predictions via predictive inference and addressing individual and interactive constraints in a sequential manner. Theoretical and numerical results demonstrate the effectiveness of the proposed method in controlling both individual and interactive constraints.

## 1 Introduction

In recent times, the field of real-time decision has flourished significantly, primarily driven by the exponential growth of available data in both the tech industry and computer science. We consider here a typical application of real-time decision, the problem of *online sample selection* [2, 5]. For instance, online recruitment systems utilize machine learning algorithms to sequentially choose qualified candidates rather than waiting for all (future) candidates' information to be collected [14]. Additionally, recommendation systems have now become commonplace in providing real-time suggestions for content (e.g., news articles, short videos)  with potential high click-through rates to users  [24]. Common situations also can be found in real-time precision marketing [43].

We describe the online sample selection problem as follows: samples (individuals) characterized by covariates $\mathbf{X}_t \in \mathbb{R}^d$ arrive sequentially while their responses $Y_t \in \mathbb{R}$ remain unobserved throughout the process. The data pairs $(\mathbf{X}, Y)$ of each time are i.i.d. random vectors. At each time point, the analyst is faced with the task of deciding whether to select the current observation based on certain predetermined criteria related to $Y_t$, and this selection process continues until a specific stopping rule is triggered. For example, $Y_t$ is the score that measures how one candidate fits a given job position in the recruitment system and the human resource agencies aim to prioritize candidates with higher $Y_t$, such as $Y_t \geq b$. Or $Y_t$ is a binary variable where $Y_t = 1/0$ means accepting or rejecting the offer, and the companies wish to find those individuals with $Y_t = 1$ based on the $X_t$.

---

[*]Equal contribution, and the first two authors are listed in alphabetical order.
[†]Correspondence to: Haojie Ren <haojieren@sjtu.edu.cn>, Changliang Zou <zoucl@nankai.edu.cn.>

38th Conference on Neural Information Processing Systems (NeurIPS 2024).

**Our Goal and Motivation.** Our goal is to sequentially select samples whose unobserved responses $Y_t$'s are in the specified target region. A natural idea is to make decisions based on the prediction value of $Y_t$ from models associated with $(\mathbf{X}, Y)$ built on some historical data. However, neglecting the uncertainty in predictions could result in numerous false decisions, i.e., selecting those samples whose true responses are beyond the specified region. To measure the selection uncertainty, some existing works reformulate sample selection as a hypothesis testing problem and focus on controlling the online false discovery rate (FDR) [16, 33]; see more discussions and literature in Section 1.2. However, in addition to quantifying statistical uncertainty, online selections often need to take into account various constraints to find informative samples in practice, for example, cost limitations, the impacts of some covariates or the diversity of candidates in the online recruitment [39]. Hence, it's necessary to explore the covariate space to satisfy these requirements. This motivates us to investigate how to efficiently implement online sample selection with statistical guarantees under various constraints.

To address this issue, we summarize common constraints into two types, i.e., *individual constraints* and *interactive constraints*. The former one is relevant to the cost of selected samples, and one typical example is the fundamental and crucial criterion, false selection rate (FSR), which quantifies the proportion of falsely selected samples and is equivalent to the well-adopted FDR. The latter constraint captures the interactive influence among selected samples and is regarded as some kind of quadratic constraint on some pairwise functions, such as the similarity or diversity among selected samples.

While the individual constraint associated with online FDR control has gained some attentions [16, 18], it alone fails to capture the nuanced pairwise relationships among different samples. To bridge this gap, we introduce interactive constraints, which are pivotal within our framework. Building upon individual constraints, the interactive criteria significantly expand the range of constraints our method can control. Integrating two distinct types of constraints into a unified framework makes it easier to create practical algorithms and ensures theoretical guarantees.

**A motivating example: candidate screening.** As an example, in recruitment, screening from the resumes arriving sequentially to determine viable candidates who can get into interview processes is an important problem in human resource management [13, 35]. In this case, one may be interested in: (1) controlling the online FSR to enhance resource efficiency [21], and (2) maintaining a desired level of candidate diversity during the screening process, thereby reducing bias [23, 44]. The individual and interactive constraints and the novel real-time sample selection procedure we propose can solve this problem precisely. See Section 4.2 for the details and more real data examples.

## 1.1 Our Contributions

In this paper, we design a novel and flexible online selection rule to effectively ensure the above two types of constraints are under control at pre-specified levels simultaneously, named as "Individual and Interactive Constrained Online Selection" (II-COS). The main idea stems from an oracle model based on the local false discovery rate (lFDR) [12, 38], which is involved in offering valid evidence on whether $Y_t$ is our interest at each time point. With some appropriately chosen evaluating functions, the II-COS procedure entails validating whether the estimates of constraints are controlled. Simulated and real-data examples clearly demonstrate the superiority of the II-COS in terms of both online individual and interactive criteria control.

To the best of our knowledge, this is the first work to systematically bridge the predictive inference and online selection procedure with various constraints. Our contributions are summarized as follows:

- Under a unified framework, the II-COS addresses how to implement online predictive selection sufficiently in consideration of both individual and interactive constraints. It is flexible to characterize the selective uncertainty and trade-off the sampling efficiency and practical limitations in the covariate space.

- Under mild conditions, we establish the theoretical guarantee that the II-COS is able to control both individual and interactive constraints simultaneously and asymptotically under one given stopping rule.

- The II-COS is model-agnostic in the sense that its implementation is applicable to any (appropriate) learning algorithms. Extensive numerical experiments indicate that the II-COS can significantly outperform existing ones while yielding effective constraints control.

## 1.2 Related Works

Our proposed method is built upon two fundamental pillars: (1) quantifying the uncertainty of response predictions using predictive inference; and (2) systematically addressing individual and interactive constraints in a sequential manner. Our work is intricately connected to the fields of predictive inference and online multiple testing. Here we briefly review literature on these two topics.

**Predictive Inference.** One key ingredient of our proposed method is predictive/conformal inference. Conformal inference [42, 34] provides a powerful and flexible tool to achieve algorithm-agnostic uncertainty quantification of predictions. Conventionally, conformal inference aims to build the prediction intervals and enjoys valid and distribution-free properties by leveraging data exchangeability [25, 3]. Taking a different but related perspective from multiple-testing, Bates et al. [7] pioneered a method to construct conformal $p$-values to detect outliers with finite-sample FDR control. Building upon this, recent advancements have improved detection power by involving more information or performing model selection [50, 26, 27, 51, 46]. The most related works are Jin and Candès [21] and Wu et al. [45], which considered a similar scenario that one would like to select some individuals of interest by controlling FDR or maximizing the diversity of selected samples in an off-line setting. Their methods are based on the conformal $p$-values or lFDR constructed with the predicted response values, respectively. Besides the fundamental difference between online and offline paradigms, our framework for characterizing various constraints poses additional challenges in how to select samples sequentially since we have multiple goals to achieve.

**Online Multiple Testing.** When only considering individual constraint as FSR control, the online sample selection can be reformulated as online multiple testing problem. Methods for online multiple testing have received much recent attention and were pioneered by Foster and Stine [16] who proposed the so-called $\alpha$-investing strategy, which was later built upon and generalized [1, 29, 30, 18, 19]. The key idea in $\alpha$-investing and its generalizations is to compare $p$-values with dynamic thresholds and gain some extra $\alpha$-wealth for each rejection. We refer to Robertson et al. [33] for a thorough overview. Those rules suffer from the "alpha-death" issue to some extent [29], which means a permanent end to decision-making when the decision threshold is too small, i.e., the online procedure stops early. This phenomenon occurs in many existing online multiple testing algorithms, as discussed in [29]. Along a different direction, Gang et al. [17] developed a new class of structure-adaptive sequential testing (SAST) rules built on the lFDR to avoid the alpha-death issue. The SAST serves as a building block for developing our II-COS procedure and can be essentially seen as a special case of ours. Later on, Ao et al. [4] reformulated online multiple testing procedure into an online knapsack problem, providing novel policies with near-optimal regret guarantees. Additionally, Xu and Ramdas [48] proposed to use e-values [41] for online multiple testing to address the dependence. However, those existing works do not take predictive inference into account and are concerned only with online error rate control without exploration of the covariate space, which may greatly hamper their applicability.

## 2 Individual and Interactive Constrained Online Selection Procedure

### 2.1 Problem Formulation

Assume there exists a historical labeled dataset as $\mathcal{D} = \{\tilde{\mathbf{X}}_i, \tilde{Y}_i\}_{i=1}^n$, where $(\tilde{\mathbf{X}}_i, \tilde{Y}_i)$'s are independent and identically distributed (i.i.d.) from $(\mathbf{X}, Y)$. A sequence of unlabeled data $\mathbf{X}_1, \mathbf{X}_2, \cdots \sim \mathbf{X} = (X_1, \cdots, X_d)^\top$ arrives in a stream with unknown responses $Y_1, Y_2, \cdots$. At each time $t$, one must make a real-time decision about whether or not to select the $t$-th individual, which is determined by some pre-specified requirement on $Y_t$. Denote $\mathcal{A}$ as the target region of $Y_t$, which differs depending on users' specifications. For example, in a regression setting, the requirement could be of the form $Y_t \in [a, b]$, $(-\infty, a)$ or $Y_t \geq b$.

Let $\theta_t = \mathbb{I}\{Y_t \in \mathcal{A}\}$ describe the true state of $Y_t$. Denote a decision rule as $\delta_t \in \{0, 1\}$, where $\delta_t = 1$ indicates that the $\mathbf{X}_t$ is selected and $\delta_t = 0$ otherwise. A false selection is made if $\delta_t = 1$ but $\theta_t = 0$. Denote $\boldsymbol{\delta}^t = \{\delta_i : i \leq t\}$ as the decision rule and $T$ as the time that the procedure stops. Our goal is to build a decision rule $\boldsymbol{\delta}^t$ to select samples with $\{Y_t \in \mathcal{A}\}$ up to stopping time $T$ such that the following two general types of constraints hold simultaneously.

**Individual Constraint.** In practice, one main concern is to control the cost of selecting samples of interest. For example, in online recruitment, companies need to control the proportion of selected

unqualified candidates or the average loss when hiring someone who rejects the offer. In such cases, we can assign each selected sample a cost associated with some pre-specified function of the covariate $X$ and control the expected cost associated with time $T$ at the target level. We refer to this requirement as *individual constraint* and write it as:

$$C_1(\boldsymbol{\delta}^t) = \mathbb{E}\left[\frac{\sum_{i\leq t}\{(1-\theta_i)G_0(\mathbf{X}_i) + \theta_i G_1(\mathbf{X}_i)\}\delta_i}{(\sum_{i\leq t}\delta_i)\vee 1}\right], \tag{1}$$

where $a\vee b = \max\{a,b\}$ and $G_0(X) \geq 0$ and $G_1(X) \geq 0$ with $G_0 \neq G_1$ are the costs corresponding to $\theta = 0$ and $\theta = 1$, respectively. Here, we take expectation due to the randomness of $\theta_1,\cdots,\theta_t$.

For example, when we simply choose $G_0(\mathbf{X}) = 1$ and $G_1(\mathbf{X}) = 0$, the individual constraint is the popular false selection rate (FSR), i.e.

$$C_1(\boldsymbol{\delta}^t) = \mathrm{FSR}(\boldsymbol{\delta}^t) = \mathbb{E}\left[\frac{\sum_{i\leq t}(1-\theta_i)\delta_i}{(\sum_{i\leq t}\delta_i)\vee 1}\right]. \tag{2}$$

The FSR is essentially equivalent to the well-adopted FDR in multiple testing literature, which is a useful tool to maintain the ability to reliably select samples of interest without excessively false selections [9]. Some works on online FDR control have been well studied. [16, 1].

The individual constraints alone cannot capture the pairwise relationship among different samples. We address this by introducing interactive constraints below.

**Interactive Constraint** Another common concern is the *interactive constraint*, which involves choosing more preferable samples. For example, companies would like to retain candidates with a diverse range of backgrounds and experiences in online recruitment, or real-time suggested contents are required to avoid homogeneity in recommendation systems. Here, we introduce a bi-variate weight function $g(\mathbf{X}, \mathbf{X}')$ to evaluate the interaction between selected samples. Denote $\mathrm{PC}(\boldsymbol{\delta}^t) = \sum_{1\leq i<j\leq t}\sum g(\mathbf{X}_i, \mathbf{X}_j)\theta_i\theta_j\delta_i\delta_j$, $\mathrm{PS}(\boldsymbol{\delta}^t) = \sum_{1\leq i<j\leq t}\sum \theta_i\theta_j\delta_i\delta_j$. We define the interactive constraint as

$$\tilde{C}_2(\boldsymbol{\delta}^t) = \mathbb{E}\left[\frac{\mathrm{PC}(\boldsymbol{\delta}^t)}{\mathrm{PS}(\boldsymbol{\delta}^t)}\right]. \tag{3}$$

Here, since only the correctly selected samples are of interest, the constraint is concerned with the average mutual effects between the correctly selected ones rather than all selected ones. When choosing the function $g$ as some similarities, controlling $\tilde{C}_2(\boldsymbol{\delta}^t)$ at a specified constant $K$, i.e., $\tilde{C}_2(\boldsymbol{\delta}^t) \leq K$, is controlling the expected similarity (ES). It is equivalent to requiring that correctly selected samples exhibit certain diversity and rich information in the covariate space of interest.

Typically, one useful choice for $g(\mathbf{X}, \mathbf{X}')$ is the weighted RBF kernel $g(\mathbf{X}, \mathbf{X}') = \exp\left\{-\frac{1}{\sigma^2}\sum_{k=1}^d w_k(X_k - X'_k)^2\right\}$ with parameter $\sigma > 0$ to measure the similarity between two independent $\mathbf{X}$ and $\mathbf{X}'$. The RBF kernel is a common and widely embraced choice in machine learning [49, 28]. Here, $\{w_1,\ldots,w_d\}$ are some given weights per users' needs. For instance, if one is just interested in the effects of the $k$-th feature, then simply $w_k = 1$ and $w_j = 0$ for $j \neq k$. Specifically, the case that $w_k = 1$ for all $k = 1,\ldots,d$ is chosen in Section 4. We also consider other similarity choices of $g(\mathbf{X}, \mathbf{X}')$, such as the cosine similarity $g(\mathbf{X}, \mathbf{X}') = \mathbf{X}^\top\mathbf{X}'/(\|\mathbf{X}\|_2\|\mathbf{X}'\|_2)$ [52].

Due to the randomness in the denominator, it turns out controlling (3) directly is not easy. Instead, we employ a modified interactive constraint,

$$C_2(\boldsymbol{\delta}^t) = \frac{\mathbb{E}\left[\mathrm{PC}(\boldsymbol{\delta}^t)\right]}{\mathbb{E}\left[\mathrm{PS}(\boldsymbol{\delta}^t)\right]}. \tag{4}$$

The constraint (4) aims to control a ratio of expectations, which is still a reasonable interactive measure. In numerical studies, we see that $\tilde{C}_2(\boldsymbol{\delta}^t)$ in (3) and $C_2(\boldsymbol{\delta}^t)$ in (4) yield almost identical patterns. An illustrative example can be found in Appendix D.1.

In sum, the goal is to select samples of interest by a decision rule $\boldsymbol{\delta}^T$ controlling both the individual and interactive constraints until stopping time $T$, i.e., $C_1(\boldsymbol{\delta}^T) \leq \alpha$ and $C_2(\boldsymbol{\delta}^T) \leq K$. We emphasize that $C_1(\boldsymbol{\delta}^T)$ and $C_2(\boldsymbol{\delta}^T)$ as well as their pre-specified levels $\alpha$ and $K$ can be chosen up to the practical applications.

## 2.2 Oracle Selection Procedure

To design a general rule that is valid for any arbitrary stopping time $T$, we consider controlling the constraints at each time $t$ in an online fashion, such that $\sup_{t \in \mathbb{N}} C_1(\boldsymbol{\delta}^t) \leq \alpha$ and $\sup_{t \in \mathbb{N}} C_2(\boldsymbol{\delta}^t) \leq K$.

Since $Y_t$ is unavailable, we consider utilizing predictive inference to measure the suspicious patterns. Let $\mu(\mathbf{x}) := Y \mid \mathbf{X} = \mathbf{x}$ be the regression or classification model associated with $(\mathbf{X}_t, Y_t)$, and one reliable estimate as $\widehat{\mu}(\cdot)$, being estimated on the labeled data $\mathcal{D}$ with some machine learning algorithm. Denote $W_t = \widehat{\mu}(\mathbf{X}_t)$ as a predicted value of $Y_t$ and assume that $\widehat{\mu}(\cdot)$ is a bijection almost surely. The bijection assumption is considerably mild and widely adopted for the identyification of each $X_t$ in the predictive inference framework [45]. The $\theta_t = \mathbb{I}(Y_t \in \mathcal{A})$ is Bernoulli$(\pi)$ distributed with $\pi = \Pr(Y_t \in \mathcal{A})$, and $W_t$ can be viewed as generated from one two-group model

$$W_t \mid \theta_t \sim (1 - \theta_t)f_0 + \theta_t f_1,$$

where $f_0$ and $f_1$ denote the probability distribution functions of $W_t$ conditional on $Y_t \notin \mathcal{A}$ (i.e., $\theta_t = 0$) and $Y_t \in \mathcal{A}$, respectively. Then, the conditional probability of $Y_t \notin \mathcal{A}$ is

$$L_t = \Pr(\theta_t = 0 \mid W_t) = \frac{(1 - \pi)f_0(W_t)}{f(W_t)}, \tag{5}$$

where $f = (1 - \pi)f_0 + \pi f_1$. The $L_t$ coincides with the local FDR in multiple testing literature [12, 17]. With the two-group model (5), we have $\mathbb{E}[\theta_t \mid \mathbf{X}_t] = 1 - L_t$ and further notice that the individual constraint $C_1(\boldsymbol{\delta}^t)$ in (1) can be exactly satisfied if

$$\frac{V_t}{R_t} := \frac{\sum_{i \leq t}\{L_i G_0(\mathbf{X}_i) + (1 - L_i)G_1(\mathbf{X}_i)\}\delta_i}{(\sum_{i \leq t} \delta_i) \vee 1} \leq \alpha,$$

holds. Here, we denote $V_t = \sum_{i \leq t}\{L_i G_0(\mathbf{X}_i) + (1 - L_i)G_1(\mathbf{X}_i)\delta_i\}$ and the number of selected ones as $R_t = \sum_{i \leq t} \delta_i \vee 1$ for notational convenience. Especially, when $G_0(\mathbf{X}) = 1$, $G_1(\mathbf{X}) = 0$, then FSR$(\boldsymbol{\delta}^t)$ in (2) can be exactly controlled.

Accordingly, the interactive constraint $C_2(\boldsymbol{\delta}^t) \leq K$ in (4) can be achieved if

$$\frac{\text{TS}_t}{\text{NS}_t} := \frac{\sum\limits_{1 \leq i < j \leq t} g(\mathbf{X}_i, \mathbf{X}_j)(1 - L_i)(1 - L_j)\delta_i\delta_j}{\sum\limits_{1 \leq i < j \leq t} (1 - L_i)(1 - L_j)\delta_i\delta_j} \leq K,$$

where the expected total mutual effects conditional on $\{\mathbf{X}_i\}_{i \leq t}$ and the expected number are denoted as $\text{TS}_t$ and $\text{NS}_t$, respectively.

Therefore, if $L_t$ is known, when a new sample $\mathbf{X}_t$ arrives at time point $t$, we can perform the decision rule as follows. Note that there is no need to consider interactive effects before the first selection. When $t$ comes before the first selection (i.e, $R_{t-1} = 0$), the decision rule is $\delta_t = 1$ if

$$\frac{V_{t-1} + L_t G_0(\mathbf{X}_t) + (1 - L_t)G_1(\mathbf{X}_t)}{R_{t-1} + 1} \leq \alpha, \tag{6}$$

holds; otherwise, $\delta_t = 0$ which means $\mathbf{X}_t$ is not selected. When $\mathbf{X}_t$ arrives with $R_{t-1} \geq 1$, then $\delta_t = 1$ if (6) and

$$\frac{\text{TS}_{t-1} + \left[\sum\limits_{i \leq t-1} g(\mathbf{X}_i, \mathbf{X}_t)(1 - L_i)\delta_i\right](1 - L_t)}{\text{NS}_{t-1} + \left[\sum\limits_{i \leq t-1} (1 - L_i)\delta_i\right](1 - L_t)} \leq K \tag{7}$$

hold simultaneously; otherwise, $\delta_t = 0$. Note that if we set $C_1(\boldsymbol{\delta}^t)$ as FSR$(\boldsymbol{\delta}^t)$ and choose $K = +\infty$, then our method essentially reduces to the same manner as the controlling step of the SAST in Gang et al. [17]. Our proposed method can be seen as a much more generalized and flexible framework for controlling both the individual and the interactive constraints simultaneously in an online fashion.

We call this method the oracle II-COS (Individual and Interactive Constrained Online Selection). The workflow in Figure 1 shows the procedure of the oracle II-COS. The following result shows that it can exactly achieve our goal.

**Proposition 2.1.** *Assume $L_t$ values are known. Then the oracle II-COS selection rule controls both constraints at any stopping time $T$, i.e., $C_1(\boldsymbol{\delta}^T) \leq \alpha$ and $C_2(\boldsymbol{\delta}^T) \leq K$.*

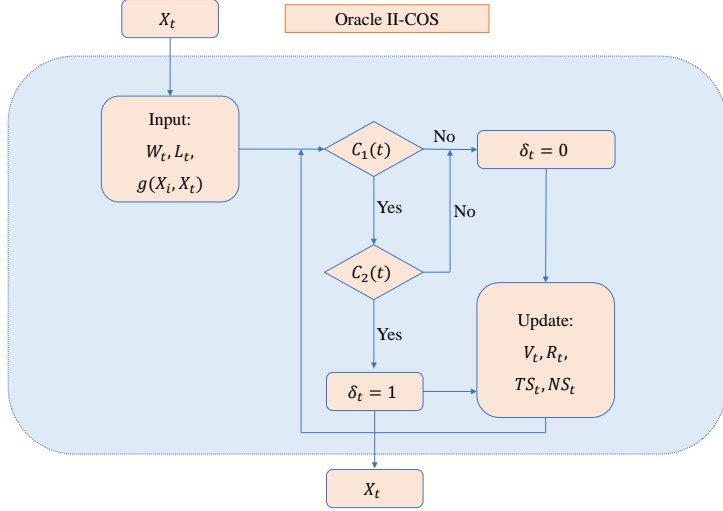

Figure 1: The implementation flowchart of the oracle II-COS procedure.

## 2.3 Data-driven II-COS Procedure

As $L_t$ is unknown in practice, we propose a data-driven II-COS procedure, which uses a reliable estimation $\widehat{L}_t$ for implementation. We resort to a data-splitting strategy: randomly split historical data $\mathcal{D}$ into two parts, the training set $\mathcal{D}_{\mathrm{tr}}$ and the calibration one $\mathcal{D}_{\mathrm{cal}}$ of sizes $n_0$ and $n_1$ respectively, where $\mathcal{D}_{\mathrm{tr}}$ is used for training a predictive model and $\mathcal{D}_{\mathrm{cal}}$ is to estimate those unknown parameters. Specifically, we first fit a regression or classification model $\widehat{\mu}(\cdot)$ on $\mathcal{D}_{\mathrm{tr}}$, and then obtain predicted values on $\mathcal{D}_{\mathrm{cal}}$, i.e. $\{\widehat{\mu}(\tilde{\mathbf{X}}_i) : (\tilde{\mathbf{X}}_i, \tilde{Y}_i) \in \mathcal{D}_{\mathrm{cal}}\}$. Note that conditional on $\mathcal{D}_{\mathrm{tr}}$, $\{\widehat{\mu}(\tilde{\mathbf{X}}_i) : (\tilde{\mathbf{X}}_i, \tilde{Y}_i) \in \mathcal{D}_{\mathrm{cal}}\}$ are i.i.d. random variables and have the same distribution as $W_t = \widehat{\mu}(\mathbf{X}_t)$, so that it can be utilized to estimate (5).

Therefore, the estimators of $f_0$ and $f$, $\widehat{f}_0$ and $\widehat{f}$, can be obtained by applying the kernel density estimation method to the data $\{\widehat{\mu}(\tilde{\mathbf{X}}_i) : (\tilde{\mathbf{X}}_i, \tilde{Y}_i) \in \mathcal{D}_{\mathrm{cal}}, \tilde{Y}_i \notin \mathcal{A}\}$ and $\{\widehat{\mu}(\tilde{\mathbf{X}}_i) : (\tilde{\mathbf{X}}_i, \tilde{Y}_i) \in \mathcal{D}_{\mathrm{cal}}\}$, respectively. And the probability $\pi = \Pr(Y_t \in \mathcal{A})$ can be approximated by $\widehat{\pi} = n_1^{-1} \sum_{(\tilde{\mathbf{X}}_i, \tilde{Y}_i) \in \mathcal{D}_{\mathrm{cal}}} \mathbb{I}(\tilde{Y}_i \in \mathcal{A})$. Further the $L_t$ in (5) can be estimated by

$$\widehat{L}_i = \frac{(1 - \widehat{\pi})\widehat{f}_0(W_i)}{\widehat{f}(W_i)} \wedge 1. \tag{8}$$

The data-driven II-COS procedure is summarized in Algorithm 1, and it indeed consists of two phases: offline estimation and online decision. The running time of offline estimation is not critical. At each time $t$, the computational complexity is a linear function of the currently selected number $R_t$. More implementation details can be found in Section 4 and Appendix B.1.

In fact, the proposed II-COS is flexible to trade off the individual and interactive constraints by adjusting the thresholds $\alpha$ and $K$. If one is concerned only with individual cost control, then we can set $K = +\infty$, with which the interactive constraint is out of work. Similarly, only the interactive effect is of interest when $\alpha = 1$. Appendix D.2 provides a toy example to illustrate this.

Before further pursuing, we would discuss the stopping time in practice. It's worth noting that the specific choice of stopping rule (and thus stopping time) is completely up to the user. For example, when $m \geq 2$ is the desired number of selections, one can set $T = \inf_t \{t : \sum_{i=1}^t \delta_i = m\}$. Or when $s$ is the total wages for recruitment, one can set $T = \inf_t \{t : \sum_{i=1}^t \delta_i s_i = s\}$ where $s_i$ is the payroll for each selected candidate. Or $T$ just chosen as one given deadline. With the use of II-COS, practitioners have the flexibility to design diverse stopping strategies that can adapt seamlessly to their specific applications. In brief, our method is flexible and is appropriate for various goals based on the user's requirements by choosing different $G_0(\mathbf{X}), G_1(\mathbf{X}), g(\mathbf{X}_i, \mathbf{X}_j)$ and varied target levels $(\alpha, K)$ and a user-specified stopping time.

---

**Algorithm 1** The data-driven II-COS procedure

---

**Input:** Target levels $\alpha$ and $K$, pairwise function $g$, cost $G_0(\mathbf{X})$ and $G_1(\mathbf{X})$, stopping time $T$, interested region $\mathcal{A}$, labeled data $\mathcal{D}$, prediction algorithm $\mathcal{H}$.

**Initialization:** $t = 0$, $V_t = R_t = 0$, $\mathrm{TS}_t = \mathrm{NS}_t = 0$; Decision rule $\boldsymbol{\delta}^t = \emptyset$

**Estimation:** Randomly split $\mathcal{D}$ into training set $\mathcal{D}_{\mathrm{tr}}$ and calibration set $\mathcal{D}_{\mathrm{cal}}$. On $\mathcal{D}_{\mathrm{tr}}$, fit $\widehat{\mu}(\mathbf{x})$ with $\mathcal{H}$. Obtain $\widehat{\pi}$, $\widehat{f}_0$ and $\widehat{f}$ from $\mathcal{D}_{\mathrm{cal}}$.

**Online decisions: while** $t \leq T$ **do**

>   Set $t = t + 1$. Compute $W_t = \widehat{\mu}(\mathbf{X}_t)$, $\widehat{L}_t$ by (8) and the similarities $g(\mathbf{X}_i, \mathbf{X}_t)$ for those $\delta_i = 1$.
>   Replace $L_t$ by $\widehat{L}_t$ in (6) and (7).
>   **if** $R_{t-1} < 1$ *and (6) holds* **then** $\delta_t = 1$;
>   **else** $\delta_t = 0$;
>   **if** $R_{t-1} \geq 1$, *(6) and (7) hold* **then** $\delta_t = 1$;
>   **else** $\delta_t = 0$;
>   Update: $\boldsymbol{\delta}^t = \boldsymbol{\delta}^{t-1} \bigcup \{\delta_t\}$; $R_t = R_{t-1} + \delta_t$; $V_t = V_{t-1} + \{\widehat{L}_t G_0(\mathbf{X}_t) + (1 - \widehat{L}_t)G_1(\mathbf{X}_t)\}\delta_t$;
>
>   $$\mathrm{TS}_t = \mathrm{TS}_{t-1} + \left[\sum_{i<t} g(\mathbf{X}_i, \mathbf{X}_t)(1 - \widehat{L}_i)\delta_i\right](1 - \widehat{L}_t)\delta_t; \quad \mathrm{NS}_t = \mathrm{NS}_{t-1} + \left[\sum_{i<t}(1 - \widehat{L}_i)\delta_i\right](1 - \widehat{L}_t)\delta_t;$$

**end**

**Output:** Selection set $\{\mathbf{X}_i : \delta_i = 1, \ \delta_i \in \boldsymbol{\delta}^T\}$.

---

**Extension to varying proportion case.** In practice, the distribution of $(\mathbf{X}_t, Y_t)$ may vary smoothly over time. In Appendix C, we consider the probability of $Y_t \in \mathcal{A}$ (i.e. the proportion of samples in the specified region) varying over time and extend the proposed II-COS to learn $\pi_t = \Pr(Y_t \in \mathcal{A})$ continuously over time and we also construct the corresponding theoretical guarantees.

## 3 Statistical Performance Guarantees

In this section, we provide statistical guarantees for the data-driven II-COS procedure. The main difficulties lie in the quantification of data-driven estimation error of $L_t$ and we utilize the classical kernel density estimation theory along with the structure of our online procedure to effectively characterize it. For simplicity, we consider that the training data set is given such that the estimated model $\widehat{\mu}$ is fixed. Before presenting our theoretical results, we state the following regularity conditions.

**Assumption 3.1** (Density functions and kernel). The density functions and kernel function satisfy

(1) The $f_1(\cdot)$ and $f_0(\cdot)$ are upper bounded by $M > 0$, and the $f(\cdot)$ is lower bounded by $\ell > 0$.

(2) The $f_0$ and $f_1$ are Hölder-continuous, i.e. $|f_0(w) - f_0(w')| \leq c_\beta |w - w'|^\beta$ for any $w, w' \in \mathbb{R}$, and the same for $f_1$ with some fixed $0 < \beta \leq 1$ and constant $c_\beta$.

(3) Kernel $K(\cdot)$ is a bounded symmetric function and enjoys exponential decay.

**Assumption 3.2** (Weight functions). There exists constants $c_G > 0$ and $c_g > 0$ such that $0 < G_0(\mathbf{X}) \leq c_G$, $0 < G_1(\mathbf{X}) \leq c_G$ for any $\mathbf{X}$ and $0 < g(\mathbf{X}, \mathbf{X}') \leq c_g$ for any $\mathbf{X} \neq \mathbf{X}'$.

Assumption 3.1 is considerably mild and widely adopted in the uniform convergence of kernel density estimation [36]. If $f_0$ and $f_1$ have bounded first-order derivatives, the Hölder-continuous assumption would hold with $\beta = 1$. The lower bound of $f$ is to ensure the uniform convergence of the estimated lFDR. Assumption 3.2 is mild since the weight functions are required only to be positive and bounded. It can be satisfied by a large category of $G_0$, $G_1$ and $g$. For example, we can take $G_j(\mathbf{X}) = a_j \|\mathbf{X}\|_2^2$ for $j = 0$ or 1 and $c_G$ exists when $\mathbf{X}$ is bounded. And we can set $g$ as the RBF and orthogonal similarities with $c_g = 1$.

With those regularity conditions, we establish the validity of the II-COS procedure for the individual constraint control.

**Theorem 3.3** (Bound for individual constraint). *Suppose Assumptions 3.1 and 3.2 hold and take the bandwidths for estimating $f$ and $f_0$ in the order of $n_1^{-1/(2\beta+1)}$. Then for any given time $t$, the individual constraint of the II-COS procedure (Algorithm 1) satisfies $C_1(\boldsymbol{\delta}^t) \leq \alpha + \Delta_{n_1}$, where $\Delta_{n_1} = D n_1^{\frac{-\beta}{2\beta+1}} \sqrt{\log n_1}$ and $D$ is a constant depending on $M$, $\ell$, $c_\beta$, $\beta$, $\pi$, $c_G$ and $K(\cdot)$.*

Although the $C_1(\cdot)$ of II-COS might be slightly larger than the target level in finite samples, this gap converges to 0 asymptotically as $n_1$ increases. In the numerical studies, we find that a small calibration size of around 200 is enough to control the $C_1$ in a reasonable range. Taking the FSR as an individual constraint, Theorem 3.3 indicates that our method can provide asymptotic online FSR control similar to an online FDR control procedure [16].

The next theorem examines the performance of the II-COS in terms of interactive constraint.

**Theorem 3.4** (Bound for interactive constraint). *Suppose Assumptions 3.1-3.2 hold and take the bandwidths for estimating $f$ and $f_0$ in the order of $n_1^{-1/(2\beta+1)}$. Let $T_s = \inf\{t : \sum_{i=1}^t \delta_i = s\}$ for $s > 2$ and assume there exists a constant $\alpha' \in (0,1)$ such that $\sum_{i \leq t} \widehat{L}_i \delta_i / (1 \vee R_t) \leq \alpha'$. Then for any given time $t \geq T_m$, the interactive constraint of the II-COS satisfies*

$$C_2(\boldsymbol{\delta}^t) \leq K + \frac{(K + c_g)\Delta_{n_1}}{0.5 - \frac{m\alpha'}{m-1} - \Delta_{n_1}}.$$

The term $m\alpha'/(m-1)$ is used to characterize the lower bound of the denominator term of the interactive constraint. Specifically, when we choose FSR as the individual constraint, we have $\alpha' = \alpha$, which demonstrates the interdependence between controlling individual and interactive constraints. Furthermore, under arbitrary stopping strategies with a stopping time $T$, we can have the asymptotic guarantee.

**Corollary 3.5.** *Suppose the conditions in Theorem 3.4 hold, the stopping moment $T \geq T_m$ and $\alpha' < (1 - 1/m)/2$. Then the II-COS procedure controls the individual and interactive constraints asymptotically at $T$, i.e. $\lim_{n_1 \to \infty} C_1(\boldsymbol{\delta}^T) \leq \alpha$ and $\lim_{n_1 \to \infty} C_2(\boldsymbol{\delta}^T) \leq K$.*

## 4 Experiments and Evaluation

We illustrate the breadth of applicability of the II-COS procedure by experiments on simulated data and real-data applications. As an example, we set the stopping rule as selecting total $m = 100$ samples, i.e., $T = T_m = \inf_t\{t : \sum_{i=1}^t \delta_i = m\}$. Additional experiments including the extended II-COS in Appendix C are shown in Appendix D.9. Code for implementing II-COS and reproducing the experiments and figures in our paper is available at `https://github.com/lulin2023/II-COS`.

**Implementation of II-COS**. To our best knowledge, online selection with uncertainty qualification has only been studied in the field of online multiple testing, which aims to control online FDR. Hence, we focus on using FSR as the individual criterion and modified ES as the interactive criterion. As $\mathbf{X}$ may be measured on scales with widely differing ranges in different dimensions, we assume that $\mathbf{X}$'s have been properly scaled in each dimension before computing $g$. We choose $g$ as the weighted RBF kernel with $\sigma = 1$, $w_k = 1$ here. Other choices for individual and interactive constraints are considered in Appendix D.5.

**Benchmarks**. We compare the II-COS procedure with four benchmarks from online multiple testing. The first one is a structure-adaptive sequential testing rule, the **SAST** [17], which is implemented with $\widehat{L}_t$. It can achieve the FSR control but ignore the interactive constraint. As mentioned earlier in Section 2, SAST can also be considered as a special case of our II-COS with $K = +\infty$. Its details are deferred to Appendix B.2. The other competitors are three well-known online FDR control algorithms **LOND** [18], **SAFFRON** [30] and **ADDIS** [40] implemented with the conformal $p$-values suggested by Bates et al. [7]. Refer more information in Appendix B.3. All the benchmarks can only control FDR, which demonstrates the flexibility of our method for different constraints.

**Performance Measures**. The empirical FSR, ES and stopping time ($T_m$) are evaluated using the average values of the false selection proportions, the similarity and the stopping time from 500 replications, respectively, where $T_m$ serves as a criterion for assessing selection efficiency.

## 4.1 Results on Synthetic Data

**Data Description.** We consider a classification model: $\mathbf{X} \mid Y = 0 \sim \mathcal{N}_4\left(\boldsymbol{\mu}_1, \mathbf{I}_4\right)$, and $\mathbf{X} \mid Y = 1 \sim \mathcal{N}_4\left(\boldsymbol{\mu}_2, \mathbf{I}_4\right)$, where $\boldsymbol{\mu}_1 = (5, 0, 0, 0)^\top$, $\boldsymbol{\mu}_2 = (0, 0, -3, -2)^\top$. We set $\Pr(Y = 1) = 0.2$. The information set is $\mathcal{A} = \{1\}$. The predictor $\mathcal{H}$ is taken as random forest with defaulted parameters. We also consider a regression setting and conduct additional experiments in Appendix D.

Firstly, we observe that methods relying on conformal $p$-values, such as LOND, SAFFRON, and ADDIS, encounter the alpha-death (stop early) issue [29]. These methods struggle to select an adequate number of samples, especially in small calibration sets. In contrast, II-COS ensures the control of both individual and interactive constraints even with a small $n_{\text{cal}}$ (e.g., 200). See more details and results in Appendix D.3. Hence, to make a fair comparison, we consider a relatively large size of the calibration set, $n_{\text{cal}} = 4,000$. We fix training data size $n_{\text{tr}} = 1,000$.

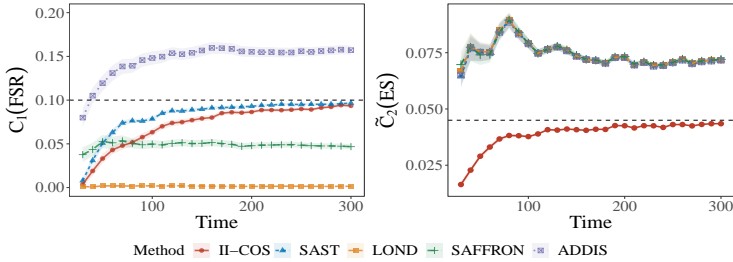

Figure 2: The values of $\text{FSR}(\boldsymbol{\delta}^t)$ and $\text{ES}(\boldsymbol{\delta}^t)$ over time $t$ for II-COS, SAST, LOND, SAFFRON and ADDIS. The black dashed lines denote the FSR level $\alpha = 0.1$ and the ES level $K = 0.045$. Shading represents error bars of one standard error above and below.

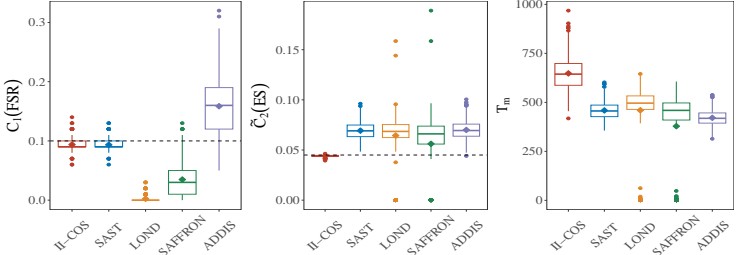

Figure 3: Boxplots of $\text{FSR}(\boldsymbol{\delta}^{T_m})$, $\text{ES}(\boldsymbol{\delta}^{T_m})$ and stopping time $T_m$ for II-COS, SAST, LOND, SAFFRON and ADDIS. The black dashed lines indicate the corresponding nominal levels.

**Results.** Figure 2 presents the online FSR (false selection rate) and ES (expected similarity) values of five methods against time $t$, which are individual and interactive constraints, respectively. The FSR levels of II-COS and SAST are closer to the nominal level than the other three methods. As expected, only the empirical ES levels of II-COS are controlled under the pre-specified level $K$ over time $t$. The LOND and SAFFRON lead to slightly conservative FSR values, while the FSR levels of ADDIS are inflated compared to the target level. Figure 3 further displays the boxplots of empirical FSR, ES at stopping time $T_m$. We observe that only II-COS achieves satisfactory ES values compared to the nominal level. Moreover, the II-COS has a relatively larger value of $T_m$ compared to those of other benchmarks. This is consistent with the fact that the II-COS spends more time exploring the structure information inside the covariate space due to the requirement of interactive constraint. Similar conclusions for the regression example in Appendix D.4 can be drawn.

Regarding efficiency, we also conducted an experiment to compare the effectiveness of II-COS with an oracle method possessing knowledge of true state $\theta_t$. The $T_m$ of II-COS is very close to the oracle. This close proximity indicates the high efficiency of II-COS. See Appendix D.8 for the details.

## 4.2 Results on Real Data

We next demonstrate the performance of the II-COS in two real-world applications. Since those online multiple testing methods based on conformal $p$-values yield few selected individuals, we focus

on comparing the II-COS with the SAST. For comparison, we also include the offline method using conformal $p$-values, where $\delta_t = 1$ if $\widehat{p}_t \le \alpha$. We denote it as CP.

**Datasets.** We consider the recruitment dataset from Kaggle [22] that contains 45,372 candidates after removing the missing data and records a binary response indicating whether the candidate passes the job interview, and other 11 attributes including education status, handicapped or not, and gender. The other problem is to use 1994 Census Bureau dataset [8] to select a subset of individuals who may have high incomes in precision marketing. This census dataset records 32,561 individuals with their 14 attributes, including gender, race, marriage, education length and so on.

For each dataset, we randomly partition the data into three parts: $n_{\mathrm{tr}} = 1,000$ training data, $n_{\mathrm{cal}} = 1,000$ calibration data and the rest which are used as the online observations. The categorical attributes are converted into one-hot codes and then are treated as numerical attributes for computing similarity measures. The prediction algorithm $\mathcal{H}$ is random forest with defaulted parameters.

**Results.** Table 1 reports the results among 500 repetitions. Both the II-COS and SAST enjoy valid FSR control, but CP yields an inflated FSR level in income investigation. The II-COS performs well in terms of similarities. To further compare the diversities, we present the proportions of different education status in in Figure 4. It can be seen that the proposed II-COS demonstrates its superior diversity in the specific attributes. See Appendix D.6 for more results for the real data. In summary, the proposed II-COS works well for selecting individuals of interest to achieve various constraints in practical applications.

Table 1: Average values with candidate dataset and income dataset: $\mathrm{FSR}(\boldsymbol{\delta}^{T_m})$, $\mathrm{ES}(\boldsymbol{\delta}^{T_m})$ ($\times 10^{-3}$) and stopping time $T_m$. The target FSR level is $\alpha = 0.2$ for both. For the candidate data, the target ES level $K = 1 \times 10^{-3}$; For the income data, $K = 6 \times 10^{-3}$. The bracket contains the standard error.

(a) Candidate dataset [22]

| Method | FSR | ES | $T_m$ |
|---|---|---|---|
| II-COS | 0.19 (0.002) | 0.98 (0.005) | 2227 (91.6) |
| SAST | 0.19 (0.002) | 8.73 (0.269) | 310 (35.2) |
| CP | 0.16 (0.002) | 10.34 (0.084) | 277 (1.49) |

(b) Income dataset [8]

| Method | FSR | ES | $T_m$ |
|---|---|---|---|
| II-COS | 0.16 (0.007) | 5.56 (0.128) | 2760 (227.0) |
| SAST | 0.19 (0.008) | 30.90 (1.078) | 1200 (202.3) |
| CP | 0.42 (0.006) | 18.84 (0.640) | 283 (3.883) |

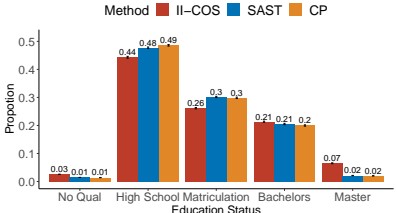 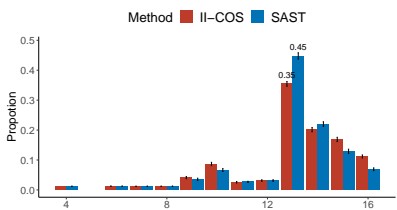

Figure 4: Left: Education status composition of the correctly selected samples (II-COS, SAST and CP) in candidate dataset; Right: Education length (year) composition of the correctly selected samples (II-COS, SAST) in income dataset. The plots have error bar to show the variation across the 500 runs.

## 5 Concluding Remarks

**Broader Impacts.** This work focuses on creating reliable machine learning tools for making real-time decisions. One key achievement is a new algorithm called II-COS, designed to select informative samples in real-time while meeting two types of general constraints. II-COS allows for both individual and interactive control, validated through theoretical analysis and numerical tests. Our method is model-agnostic and easily applicable to many real-world cases such as producing diversified results while controlling FSR for online recruitment. One potential negative impact of our work is that researchers will apply the algorithm without sufficient scrutiny. We emphasize that it's important to use caution when applying this method to complex real-world scenarios to prevent misuse.

**Limitations.** Firstly, we mainly consider binary functions as the interactive constraint. How to adapt the II-COS to other popular constraints, such as the Gini index, deserves further study. Secondly, in certain practical scenarios, it is possible to obtain feedback after decisions. Incorporating the feedback information into our method to enhance its performance warrants future research.

## Acknowledgments and Disclosure of Funding

We thank anonymous area chair and reviewers for their helpful comments. Zou was supported by the National Key R&D Program of China (Grant Nos. 2022YFA1003703, 2022YFA1003800), the National Natural Science Foundation of China (Grant Nos. 11925106, 12231011, 11931001, 12226007, 12326325). Ren was supported by the National Natural Science Foundation of China (Grant Nos. 12101398, 12471262), and Young Elite Scientists Sponsorship Program by CAST.

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

# Supplementary Material for "Real-Time Selection Under General Constraints via Predictive Inference"

This supplementary material contains:

- Preliminary terms for self-containment (Appendix A);

- Implementation details (Appendix B);

- An extension algorithm to varying proportion case (Appendix C);

- Additional experiments (Appendix D);

- The proofs of all the theoretical results. (Appendix E).

## A   Preliminary Terms for Self-Containment

Here, we list the preliminary terms we use in the paper for the sake of clarity and self-containment.

- FDR [9], false discovery rate, a widely-adopted error rate notion in the field of multiple testing, is defined as the expected proportion of incorrectly rejected null hypotheses as follows:

$$\text{FDR}(t) = \mathbb{E}\left[\frac{|\mathcal{H}_0 \cap \mathcal{R}(t)|}{|\mathcal{R}(t)| \vee 1}\right],$$

where $\mathcal{H}_0$ is the unknown set of true null hypotheses, $\mathcal{R}(t)$ represents the set of rejected null hypotheses until time $t$ and then $\mathcal{H}_0 \cap \mathcal{R}(t)$ is the set of false discoveries.

- FSR, false selection rate, defined as the expected proportion of individuals being not of interest among the selected subset of individuals. It is in fact equivalent to the definition of FDR. In our framework in this paper, we describe it equivalently as:

$$\text{FSR}(t) = \mathbb{E}\left[\frac{\sum_{i \leq t}(1 - \theta_i)\delta_i}{(\sum_{i \leq t}\delta_i) \vee 1}\right].$$

## B   Implementation Details of Algorithms

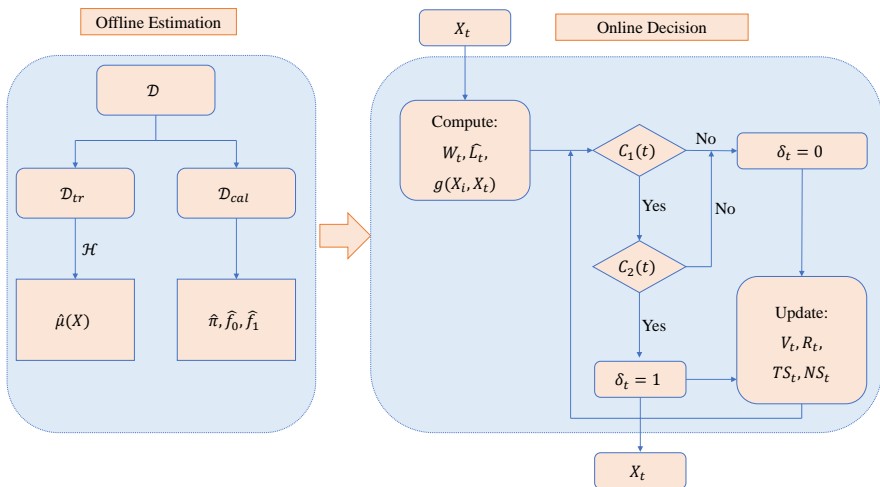

Figure 5: The implementation flowchart of the data-driven II-COS procedure.

## B.1 Implementation Details of II-COS

Figure 5 shows the implementation flowchart of our proposed data-driven II-COS procedure, including the offline estimation step and online decision step. Here we introduce more implementation details to reproduce the results.

**About the classifier.** For the classification problem, we use the predicted conditional probability $\widehat{\Pr}(Y_t = 1 \mid \mathbf{X}_t = \mathbf{x})$ as $\widehat{\mu}(\mathbf{x})$ in the procedure of II-COS. Many commonly used algorithms such as random forest and neural networks can provide such probability estimators. In fact, the choice of $\widehat{\mu}$ is not restricted to probability. For example, the support vector machine outputs the distance to the separating hyperplane for classification, and such distance can be chosen as $\widehat{\mu}$.

However, for probability estimators, one potential problem is that when the classifier is quite accurate, most of $W_i$'s are concentrated at $0$ or $1$ and few of them take values between $(0, 1)$. In such a situation, it is difficult to accurately and stably estimate lFDR $L_i$'s since the density functions $f_0(w)$ and $f(w)$ are not smooth enough and are not lower bounded. This yields that our assumptions are violated, but the II-COS can still perform satisfactorily with some corrections on the estimation $\widehat{L}_i$'s. Notice that the larger $W_i$ is, the more likely $Y_i \in \mathcal{A}$ and hence the smaller $L_i$ is. That is $\widehat{L}_i$ should be monotonically decreasing as $W_i$ increases. Observing this, we can make a monotonization correction on $\widehat{L}_i$'s. To be specific, we rearrange $\widehat{L}_i$ by the decreasing order of $W_i$. If $\widehat{L}(W_{(i-1)}) < \widehat{L}(W_{(i)})$ which violates the monotonicity, we revise it by $\widehat{L}(W_{(i-1)}) = \widehat{L}(W_{(i)})$, where $W_{(i)}$ is the $i$th smallest among $W$. This monotonization correction enables us to avoid obvious errors due to unstable estimation and improve the performances of all the methods utilizing $\widehat{L}_i$'s.

**Choice of $K$.** A useful interactive constraint needs an appropriate specification of $K$. For any two i.i.d. observations $\mathbf{X}$ and $\mathbf{X}'$ with corresponding $\theta$ and $\theta'$ respectively, the expected $C_2$ of the individuals of interest is given as $C_2 := \mathbb{E}[g(\mathbf{X}, \mathbf{X}') \mid \theta = 1, \theta' = 1]$, which can be estimated by $\widehat{C}_2 = \sum\sum_{i<j; i,j \in \mathcal{L}} g(\tilde{\mathbf{X}}_i, \tilde{\mathbf{X}}_j) / \{|\mathcal{L}|(|\mathcal{L}| - 1)\}$, where $\mathcal{L} = \{i : \tilde{Y}_i \in \mathcal{A}\}$. It is reasonable to set $K = a\widehat{C}_2$, where $a > 0$ is user-specific to control the interactive constraint level. Our numerical evidence reveals that $a \in (0.1, 0.5)$ works generally well.

## B.2 Implementation Details of SAST

Gang et al. [17] proposed a structure-adaptive sequential testing (SAST) rule for online false discovery rate control. In their work, the rejecting rule is as follows: If $L_t < \gamma_t$ and $\{|\mathcal{R}_{t-1}| + 1\}^{-1} \left( \sum_{i \in \mathcal{R}_{t-1}} L_i + L_t \right) \leq \alpha$, then $\delta_t = 1$. Otherwise $\delta_t = 0$, where $\mathcal{R}_{t-1} = \{i \leq t-1 : \delta_i = 1\}$ and $\gamma_t$ is a barrier estimated from an "offline" procedure.

The implementation details of SAST for comparisons in our simulations in Section 4 are different from the original one. Firstly, the original SAST assumes the null density function $f_0$ is already known while in our setting $f_0$ remains unknown. Secondly, in our predictive inference setting, the density functions and the null proportion are directly estimated via calibration set as the offline estimation procedure in Algorithm 1, not from current rejection sets. Besides, considering the time-varying structures of the data stream in their setting, Gang et al. [17] incorporated the barrier strategy in their method, which is not necessary to be adopted here.

## B.3 Implementation Details of Conformal $p$-values

The notion of conformal $p$-values was originally proposed by Vovk et al. [42] to construct prediction interval. Recently, there exist some works to apply conformal $p$-values to implement sample selection from a multiple-testing perspective, such as Bates et al. [7], Rava et al. [31] and Jin and Candès [21]. In the sample selection problem, the hypothesis has the following form for each $t$,

$$H_{0t} : Y_t \in \mathcal{A}^c \quad \text{v.s.} \quad H_{1t} : Y_t \in \mathcal{A}.$$

There are two types of conformal $p$-values and we adopt the one in Bates et al. [7] which utilizes the same class calibration. Recall that for $W_t$, its conformal $p$-value $p_t$ is defined as

$$\widehat{p}_t = \frac{1 + \sum_{i \in \mathcal{D}_{\text{cal}}} \mathbb{I}\{Q(\tilde{W}_i) \leq Q(W_t)\}}{1 + |\mathcal{D}_{\text{cal}}|}.$$

The nonconformity score function $Q(W_t)$ is used to indicate the possibility of $\theta_t = 0$. For example, in regression settings, if $\mathcal{A} = [b, +\infty)$, we can use $Q(W_j) = b - W_j$. If $\mathcal{A} = (-\infty, a] \cup [b, +\infty)$, then we can choose $Q(W_t) = \max\{W_t - a, b - W_t\}$. And in binary classification settings, if $\mathcal{A} = 1$ and $W_t$ indicates the probability of $Y_t = 1$, we set $Q(W_t) = 1 - W_t$.

Even though conformal $p$-values are correlated, using conformal $p$-values to conduct multiple testing can control the FDR level with finite sample guarantee since they are positive regression dependent on a subset (PRDS) [10]. However, the conformal $p$-values are lower bounded by $1/(|\mathcal{D}_{\text{cal}}| + 1)$, which leads to unsatisfactory performance for online multiple testing methods based on $p$-values. Since these methods require sufficiently small $p$-values to make rejections.

In our simulations, we implement LOND, SAFFRON and ADDIS for online sample selection by R package `OnlineFDR` [32] with $\alpha = 0.1$. Other parameters are defaulted. Here we introduce the details about these online FDR control methods. Ramdas et al. [29] proposed a "statistical perspective" to control FDR in online setting, which is to keep an estimate of the FDP less than $\alpha$ similar to the offline setting. Specifically, for offline FDR, let the rejection set $\mathcal{R}(s) = \{i | p_i \leq s\}$. An oracle estimate for FDP is given by $\text{FDP}^*(s) := \frac{|\mathcal{H}_0| \cdot s}{|\mathcal{R}(s)| \vee 1}$. For online FDR, an oracle estimate of $\text{FDP}^*(t)$ is $\frac{\sum_{j \leq t, j \in \mathcal{H}_0} \alpha_j}{R(t) \vee 1}$. Table 2 lists a comparison of estimating FDP in classical offline methods and online methods for FDR control in multiple testing. For the online methods, denote the decision rule as $\delta_t = \{p_t \leq \alpha_t\}$, where $p_t$ is the corresponding conformal $p$-value at time $t$ for our problem. The test levels $\{\alpha_t\}$ for LOND [18], SAFFRON [30] and ADDIS [40] are listed as follows:

Table 2: A comparison of $\widehat{\text{FDP}}$ in offline methods v.s. online methods for FDR control.

| Offline | $\widehat{\text{FDP}}$ | $\widehat{\text{FDP}}(t)$ | Online |
|---------|------------------------|---------------------------|--------|
| BH [9] | $\frac{n \cdot s}{|\mathcal{R}(s)| \vee 1}$ | $\frac{\sum_{j \leq t} \alpha_j}{R(t) \vee 1}$ | LOND [18] |
| Storey-BH [37] | $\frac{n \cdot s \cdot \hat{\pi}_0}{|\mathcal{R}(s)| \vee 1}, \hat{\pi}_0 = \frac{\sum_{i=1}^{n} \mathbf{1}(p_i > \lambda)}{n(1-\lambda)}$ | $\frac{\sum_{j \leq t} \alpha_j \frac{\mathbf{1}\{p_j > \lambda_j\}}{(1-\lambda_j)}}{R(t) \vee 1}$ | SAFFRON [30] |
| | | $\frac{\sum_{j \leq t} \alpha_j \frac{\mathbf{1}\{\lambda_j < p_j \leq \tau_j\}}{\tau_j - \lambda_j}}{R(t) \vee 1}$ | ADDIS [40] |

- LOND: $\alpha_t = \gamma_t(R(t-1)+1)$, where $\{\gamma_t\}_{t=1}^{\infty}$ is a given infinite non-increasing sequence of positive constants that sums to $\alpha$ and $R(n) = \sum_{t=1}^{n} R_t$ denotes the number of discoveries in the first $n$ hypotheses tested.

- SAFFRON: At each time $t$, define $C_{j+} = C_{j+}(t) = \sum_{i=\tau_j+1}^{t-1}$, where $C_t = \mathbb{I}\{p_t \leq \lambda\}$. For $t = 1$, $\alpha_1 = \min\{\gamma_1 W_0, \lambda\}$; For $t = 2, 3, \ldots, \alpha_t := \min\{\lambda, \tilde{\alpha}_t\}$, where

$$\tilde{\alpha}_t = W_0 \gamma_{t-C_{0+}} + ((1-\lambda)\alpha - W_0)\gamma_{t-\tau_1-C_{1+}} + (1-\lambda)\alpha \sum_{j \geq 2} \gamma_{t-\tau_j-C_{j+}}.$$

- ADDIS: The testing levels for ADDIS are given by $\alpha_t = \min\{\lambda, \hat{\alpha}_t\}$, where

$$\hat{\alpha}_t = (\eta - \lambda)[\omega_0 \gamma_{S^t-C_{0+}} + (\alpha - \omega_0)\gamma_{S^t-\tau_1^*-C_{1+}} + \alpha \sum_{j \geq 2} \gamma_{S^t-\tau_j^*-C_{j+}}]$$

and $S^t = \sum_{i<t} \mathbb{I}\{p_i \leq \eta\}$, $\tau_j^* = \sum_{i \leq \tau_j} \mathbb{I}\{p_i \leq \eta\}$.

## B.4 Experiments Compute Resources

All the experiments were conducted on 3.11 GHz Intel Gen i5-11300H processors with 16 Gb memory at a Lenovo personal computer and the R platform with version 4.2.1. The time of execution for each of the individual experimental runs is about 6.686 seconds. And the total compute time for the synthetic classification example in Section 4 for 500 replications is about 63.877 minutes.

### B.5 A Toy Example for Illustration in Section 2.3

We illustrate the idea of the II-COS procedure via a binary classification example. We aim to select $m = 50$ data points of a specific class from unlabeled data arriving sequentially. We choose FSR as the individual constraint and mES as the interactive constraint.

The data is generated as follows. The 4-dimensional covariates $\mathbf{X} = (X_1, X_2, X_3, X_4)^\top$ are generated from a mixture of multivariate normal distributions with mean $(0, 0, 0, 0)$ if $Y = 0$ and mean $(-3, -3, 0, 0)$ if $Y = 1$. The proportions of $Y = 0$ and $Y = 1$ are 80% and 20% respectively. In this illustrative example, we set the historical data size $n = 1,000$ with calibration size $n_1 = 500$ and would like to select $m = 50$ data points from the interest region $\mathcal{A} = \{1\}$ with the target FSR level $\alpha = 0.1$. The mES threshold $K$ is set at $0.01$ for II-COS.

Figure 6 depicts the scatterplot of the first two dimensions of the covariates $\mathbf{X}$, with green dots and red triangles denoting correctly selected points and falsely selected ones, respectively. The SAST method proposed by Gang et al. [17] is taken as one benchmark, which considers only the FSR control. We observe that the selected points of II-COS enjoy significant diversity among the covariate space and only a few false selections are contained. In contrast, the SAST is inclined to choose similar samples concentrated at the center of the concerned group and stops too early to fully explore the covariate space with sequentially arriving samples.

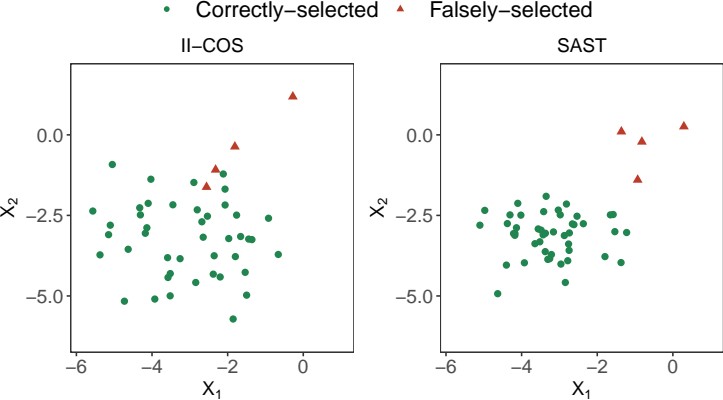

Figure 6: Scatter plots of selected points of the II-COS and SAST. It stops when selecting 50 samples . Left: the selection results of the II-COS; Right: the results of SAST. Green dots and red triangles indicate correct and false selections, respectively.

## C  Extension to Varying Proportion Case

In practice, the distribution of $(\mathbf{X}_t, Y_t)$ may vary smoothly over time. Due to the unknown $Y_t$, it is unrealistic to re-estimate parameters (i.e., **Estimation Step** in Algorithm 1) on both labeled data and the most recent data. To mitigate this problem, we consider the probability of $Y_t \in \mathcal{A}$ (i.e. the proportion of samples in the specified region) varying over time and extend the proposed II-COS to learn $\pi_t = \Pr(Y_t \in \mathcal{A})$ continuously over time.

Let $Q(W_t)$ be one score function, which is high when the possibility of $\theta_t = 0$ is large and otherwise low. For example, in regression settings, if $\mathcal{A} = [a, +\infty)$, we can take $Q(W_t) = a - W_t$. And in binary classification, if $\mathcal{A} = 1$ and $W_t$ indicates the probability of $Y_t = 1$, we can set $Q(W_t) = 1 - W_t$. One valid conformal $p$-value for $Q(W_t)$ can be obtained as Bates et al. [7],

$$\widehat{p}_t = \frac{1 + \sum_{i \in \mathcal{D}_{\text{cal}}} \mathbb{I}\{Q(\tilde{W}_i) \leq Q(W_t)\}}{1 + |\mathcal{D}_{\text{cal}}|}.$$

Inspired by the techniques for null proportion estimation in multiple testing literature [37], we note that $\Pr(\widehat{p}_t \geq \lambda) \approx \Pr(\widehat{p}_t \geq \lambda, \theta_t = 0) \approx \pi_t(1 - \lambda)$ for large $\lambda \in (0, 1)$ (i.e., $\lambda = 0.5$).

Thus, we consider using some recent $\widehat{p}_t$ to estimate a reliable $\pi_t$. Denote $q$ as the size of a neighborhood $\{t - q, \ldots, t - 1\}$ and fix $\lambda$. We employ an exponential weighted scheme to estimate $\pi_t$ where

the more recent samples will contribute more to the estimation:

$$\widehat{\pi}_t^\lambda = 1 - \frac{\sum_{j=t-q}^{t-1} \kappa_b(j-t)\mathbb{I}\{\widehat{p}_j > \lambda\}}{(1-\lambda)\sum_{j=t-q}^{t-1}\kappa_b(j-t)},$$

where $\kappa_b(s) = \exp\{-|s|/b\}$ and $b$ is the bandwidth parameter. Then, we can compute the distribution of $W_t$ as $\widehat{f}^t = \widehat{f}_0(w)(1-\widehat{\pi}_t) + \widehat{f}_1(w)\widehat{\pi}_t$ and estimate $L_t$ by $\widehat{L}_t^\lambda = \frac{(1-\widehat{\pi}_t^\lambda)\widehat{f}_0(W_t)}{\widehat{f}^t(W_t)} \wedge 1$.

The extended Π-COS is to substitute $\widehat{L}_t^\lambda$ for $\widehat{L}_t$ in Algorithm 1. The following theorem establishes the guarantee by assuming the slow change of distribution [6].

**Theorem C.1.** *Assume $\pi_t$ satisfies $|\pi_{t+1} - \pi_t| \leq \eta$ for any $t$, and the bandwidth parameter $b$ satisfies $b^\zeta \leq q$ for some $\zeta > 1$. Denote $\Delta' = c_1 n_1^{-\beta/(2\beta+1)}\sqrt{\log n_1} + c_2\max\{b^{-1/3}, (\log n_1/n_1)^{1/6}, (b\eta)^{2/3}\}$ with constants $c_1, c_2$. Suppose $p_t$ and $\lambda$ satisfy $\Pr(p_t > \lambda \mid \theta_t = 1) = 0$. Under Assumption 3.1-3.2, we have:*

*(a) For any given time $t$, the individual constraint of extended Π-COS satisfies $C_1(\boldsymbol{\delta}^t) \leq \alpha + \Delta'$;*

*(b) Furthermore, if the conditions in Theorem 3.4 hold, then for any given time $t > T_m$, the interactive constraint of the extended Π-COS satisfies $C_2(\boldsymbol{\delta}^t) \leq K + \frac{(K+c_g)\Delta'}{0.5 - \frac{m\alpha'}{m-1} - \Delta'}$.*

Besides the part similar to $\Delta_{n_1}$ in Theorem 3.3, the bound $\Delta'$ owns an additional term that can be decomposed into three parts. The $b^{-1/3}$ indicates how many valid samples we use to estimate $\pi_t$. The second part comes from the approximation error of $p$-values. The last one characterizes the effects of distribution shift. If we properly choose $b$ such that $b\eta = o(1)$, this term is negligible. Thus when $b$ and $n_1$ both tend to infinity, $\Delta'$ converges to 0 and the individual and interactive constraints will be controlled asymptotically.

# D  Additional Experiments

## D.1  Illustration of the Similar Patterns between $\tilde{C}_2$ and $C_2$

We calculate the empirical $\tilde{C}_2$ and $C_2$ during the online sample selection procedure under regression setting in Section D.4 with Π-COS from 500 replications. The results are summarized in Figure 7.

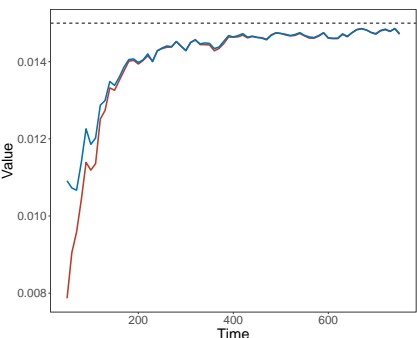

Figure 7: Illustration of the similar patterns between $\tilde{C}_2$ and $C_2$. Line charts of $\tilde{C}_2(\boldsymbol{\delta}^t)$ (red line) and $C_2(\boldsymbol{\delta}^t)$ (blue line) over time $t$ for the regression example. Experiment details are in Appendix D.4. The black dashed line is $K = 0.015$. The two measures yield almost identical patterns.

## D.2  An Illustration Example of the Flexibility for Choices of $\alpha$ and $K$.

There exists some trade-off of stopping time and two criteria. In fact, Π-COS could result in a short stopping time when only one criterion is considered. Typically, one could choose $K = +\infty$ for the case the interactive constraint is out of work and only individual criterion control is considered, and meanwhile one can set $\alpha = 1$ with which the interactive constraint is the only concern. The results in Table 3 evaluated this conclusion where $C_1$ is the FSR and $\tilde{C}_2$ is the ES.

Table 3: Results of flexible choice of $\alpha$ and $K$ for the classification example. Average values among 500 repetitions: $\text{FSR}(\boldsymbol{\delta}^{T_m})$, $\text{ES}(\boldsymbol{\delta}^{T_m})$ and stopping time $T_m$.

| $\alpha, K$ | $C_1(\text{FSR})$ | $\tilde{C}_2(\text{ES})$ | $T_m$ |
|---|---|---|---|
| $\alpha = 0.1, K = +\infty$ | 0.101 | 0.073 | 446.51 |
| $\alpha = 0.1, K = 0.045$ | 0.102 | 0.044 | 628.82 |
| $\alpha = 1, K = 0.045$ | 0.849 | 0.038 | 107.79 |

Table 4: Average number of selected samples when stopping for II-COS, LOND, SAFFRON and ADDIS.

| $n_{\text{cal}}$ | Classification | | | | Regression | | | |
|---|---|---|---|---|---|---|---|---|
| | II-COS | LOND | SAFFRON | ADDIS | II-COS | LOND | SAFFRON | ADDIS |
| 500 | 100 | 0.20 | 0.70 | 100 | 100 | 0.20 | 21.82 | 24.05 |
| 1000 | 100 | 0.72 | 10.40 | 100 | 100 | 0.87 | 56.72 | 59.20 |
| 1500 | 100 | 11.40 | 36.33 | 100 | 100 | 12.28 | 74.70 | 80.05 |
| 2000 | 100 | 44.20 | 50.79 | 100 | 100 | 40.99 | 80.03 | 82.02 |
| 2500 | 100 | 68.33 | 60.95 | 100 | 100 | 59.72 | 92.01 | 91.04 |

## D.3 Effects of the Calibration Size

We perform some additional simulations under the classification setting, to study the stopping early issue for those methods based on the conformal $p$-values.

Take $\alpha = 0.1$ and $K = 0.045$ for FSR and mES, respectively. In Table 4, we first fix $n_{\text{tr}} = 1,000$ and $m = 100$ and vary $n_{\text{cal}}$ from 500 to 2,500 to compare the average numbers of selected samples of these three methods with II-COS until stopping. It is clear that all the three benchmarks are unable to select enough samples across all the settings, especially with a small calibration set. As the calibration size $n_1$ increases, their selected numbers tend to be close to the target. This can be understood because a larger calibration size would generally yield more accurate detection of the individual of interest and thus alleviate the alpha-death issue to some extent. In contrast, the performances of II-COS, in terms of the number of selected samples until stopping, would be much less influenced by the size of calibration data. The II-COS only stops when $m$ samples are obtained under all the scenarios.

Furthermore, to verify that II-COS can guarantee both $\text{FSR}(\boldsymbol{\delta}^{T_m})$ and $\text{ES}(\boldsymbol{\delta}^{T_m})$ control with a relatively small calibration size $n_{\text{cal}}$ (such as 200), we apply II-COS in synthetic data. The details of the data generation process can be found in Section 4 and D.4.

We fix $n_{\text{tr}} = 1000$, $m = 100$ and $n_{\text{cal}}$ varies from 200 to 800. The average of $\text{FSR}(\boldsymbol{\delta}^{T_m})$ and $\text{ES}(\boldsymbol{\delta}^{T_m})$ with II-COS for both scenarios are calculated from 500 replications. The results are summarized as Table 5. It's obvious that for different $n_{\text{cal}}$, both $\text{FSR}(\boldsymbol{\delta}^{T_m})$ and $\text{ES}(\boldsymbol{\delta}^{T_m})$ are controlled under the pre-specified constant $\alpha$ and $K$ respectively under all scenarios with our II-COS method.

Table 5: Average values of $\text{FSR}(\delta^{T_m})(\%)$ and $\text{ES}(\delta^{T_m})(\times 10^{-2})$ (with standard errors in parentheses) for II-COS with different $n_{\text{cal}}$ under classification setting ($\alpha = 10\%, K = 4.50 \times 10^{-2}$) and regression setting ($\alpha = 10\%, K = 1.50 \times 10^{-2}$).

| $n_{\text{cal}}$ | Classification | | Regression | |
|---|---|---|---|---|
| | FSR $(T_m)$ | ES $(T_m)$ | FSR $(T_m)$ | ES $(T_m)$ |
| 200 | 9.22(0.73) | 4.41(0.08) | 5.74(0.18) | 1.49(0.03) |
| 400 | 9.22(0.69) | 4.42(0.07) | 4.84(0.19) | 1.49(0.04) |
| 600 | 9.47(0.80) | 4.39(0.09) | 5.18(0.19) | 1.49(0.04) |
| 800 | 9.36(0.69) | 4.40(0.08) | 5.79(0.19) | 1.49(0.04) |

## D.4 Results on Synthetic Data for Regression Setting

The following regression setting is considered: $Y = -7X_1^2 + 5\exp X_2 + 10(X_3 + X_4)^2 + \varepsilon$, with $\mathbf{X} \sim \mathcal{N}_4(\mathbf{0}, \mathbf{I}_4)$ and $\varepsilon \sim \mathcal{N}(0, 1)$. The informative set is $\mathcal{A} = (c, \infty)$, where $c$ is the 80% quantile of $Y$. The FSR and mES are considered as $C_1$ and $C_2$, respectively. The prediction algorithm $\mathcal{H}$ is taken as neural network, with a single hidden layer and 10 hidden neurons, implemented by R package `nnet`, and $K$ is chosen as $0.015$.

The simulation results are summarized in Figure 8 and 9 from 500 replications. The results are similar to those for classification setting. It's further verified that our proposed II-COS outperforms all the benchmarks under both classification and regression scenarios.

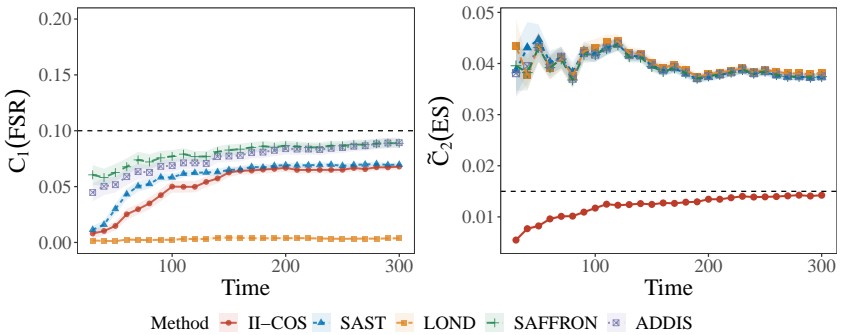

Figure 8: Simulation results for regression setting. Line charts of $\mathrm{FSR}(\boldsymbol{\delta}^t)$ (Left) and $\mathrm{ES}(\boldsymbol{\delta}^t)$ (Right) for II-COS, SAST, LOND, SAFFRON and ADDIS over time $t$. The black dashed lines are the corresponding FSR level $\alpha = 0.1$ and the ES level $K = 0.015$. Shading represents error bars of one standard error above and below.

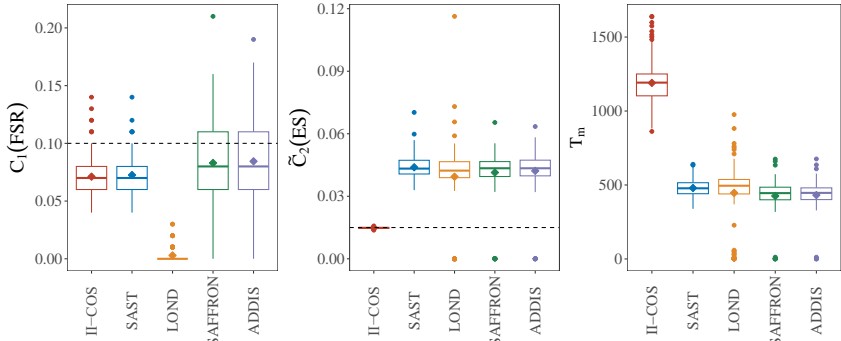

Figure 9: Simulation results for regression setting. Boxplots of $\mathrm{FSR}(\boldsymbol{\delta}^{T_m})$ (Left), $\mathrm{ES}(\boldsymbol{\delta}^{T_m})$ (Middle) and $T_m$ (Right) for II-COS, SAST, LOND, SAFFRON and ADDIS. The black dashed lines are the corresponding FSR level $\alpha = 0.1$ and ES level $K = 0.015$.

## D.5 Experiments for Other Individual and Interactive Constrains.

**Other individual constraints** To better illustrate the performance of our proposed method, we also conduct an experiment for the general individual constraint. We choose $G_0(\mathbf{X}) = |\sum_{i=1}^d X_i|$, $G_1(X) = |\sum_{i=1}^d X_i|/2$ for EC (expected cost), and choose $G_0(\mathbf{X}) = 1$, $G_1(\mathbf{X}) = 0$ for FSR. Other settings are the same as the classification model. The results are shown in Figure 10. We can see that only the proposed II-COD can guarantee all EC (expected cost), FSR and ES control, while all the benchmarks are out of control for EC or ES.

**Other pairwise function $g$.** Besides the RBF kernel, popular choices include the cosine similarity [47, 52] with adjustment $g(\mathbf{X}, \mathbf{X}') = \mathbf{X}^\top \mathbf{X}'/(\|\mathbf{X}\|_2 \|\mathbf{X}'\|_2) + 1$, and the absolute value of cosine similarity, i.e., $g(\mathbf{X}, \mathbf{X}') = |\mathbf{X}^\top \mathbf{X}'|/(\|\mathbf{X}\|_2 \|\mathbf{X}'\|_2)$ which characterizes the orthogonality between

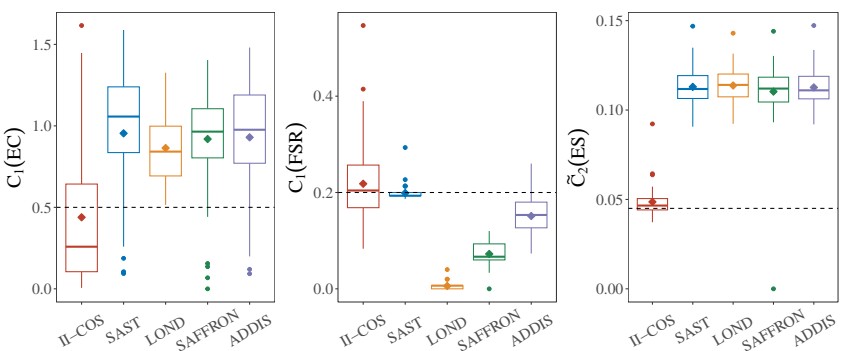

Figure 10: Boxplots of $EC(\delta^{T_m})$, $FSR(\delta^{T_m})$, and $ES(\delta^{T_m})$ for II-COS, SAST, LOND, SAFFRON and ADDIS under classification model. The black dashed lines indicate the corresponding nominal levels $\alpha_1 = 0.5, \alpha_2 = 0.2, K = 0.045$.

$\mathbf{X}$ and $\mathbf{X}'$ and is often considered in the field of design of experiments [15]. The similarity functions mentioned above all satisfy Assumption 3.2. As a supplement to the experiments in the main text, we choose the cosine similarity with adjustment $g(\mathbf{X}, \mathbf{X}') = \mathbf{X}^\top \mathbf{X}'/(\|\mathbf{X}\|_2 \|\mathbf{X}'\|_2) + 1$ for classification setting. Notice that the original cosine similarity $\mathbf{X}^\top \mathbf{X}'/(\|\mathbf{X}\|_2 \|\mathbf{X}'\|_2)$ can be negative sometimes. Hence we add a constant 1 which guarantees $g(\mathbf{X}, \mathbf{X}') \geq 0$ and does not change the final results. The simulation results are summarized in Figure 11 from 500 replications. As can be seen, the simulation results are similar to those when choosing the RBF kernel in Section 4.1.

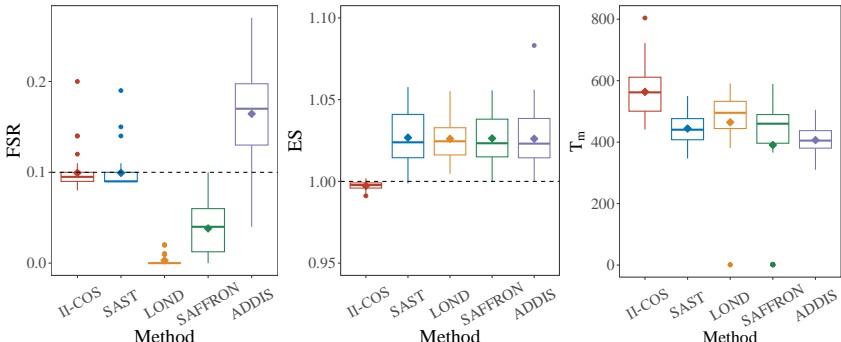

Figure 11: Boxplots of $FSR(T_m)$, $ES(T_m)$ and $T_m$ with random forest algorithms under classification setting for II-COS, SAST, LOND, SAFFRON and ADDIS. The similarity function $g$ is chosen as the cosine similarity with adjustment $g(\mathbf{X}, \mathbf{X}') = \mathbf{X}^\top \mathbf{X}'/(\|\mathbf{X}\|_2 \|\mathbf{X}'\|_2) + 1$. The black dashed lines are the corresponding FSR level $\alpha = 0.1$ and the ES level $K = 1.0$.

### D.6 Additional Results for Real Data Analysis

In Table 6, the average proportions of handicapped and female in the selected candidates are in the last two columns of (a). The average proportions of minority and female among the correctly selected individuals are in the last two columns of (b).

### D.7 Additional Experiments for Different Learning Algorithms

Except for the random forest algorithm, we further apply two different learning algorithms $\mathcal{H}$ for classification setting to estimate the model $\hat{\mu}(\mathbf{X})$: Support vector machine (SVM) and Neural network (NN) with a single hidden layer and 10 hidden neurons, which are implemented by R packages `kernlab` and `nnet`, respectively. For NN algorithm, the entropy fitting is used for classification setting. The empirical $FSR(T_m)$ and $ES(T_m)$ levels are estimated by the average of the false selection proportion and the expected similarity respectively from 500 replications. The results are summarized in Figure 12. As can be seen, the simulation results are similar as those when applying random forest in Section 4.1.

Table 6: Average values with candidate dataset and income dataset: $\text{FSR}(\boldsymbol{\delta}^{T_m})$, $\text{ES}(\boldsymbol{\delta}^{T_m})$ ($\times 10^{-3}$) and stopping time $T_m$. The average proportions of handicapped and female in the selected candidates are in the last two columns of (a). The average proportions of minority and female among the correctly selected individuals are in the last two columns of (b). The target FSR level is $\alpha = 0.2$ for both. For the candidate data, the target ES level $K = 1 \times 10^{-3}$; For the income data, $K = 6 \times 10^{-3}$.

(a) Candidate dataset [22]

| Method | FSR | ES | $T_m$ | Handicap | Female |
|---|---|---|---|---|---|
| II-COS | 0.19 | 0.98 | 2227 | 0.15 | 0.48 |
| SAST | 0.19 | 8.73 | 310 | 0.05 | 0.47 |
| CP | 0.16 | 10.34 | 277 | 0.04 | 0.46 |

(b) Income dataset [8]

| Method | FSR | ES | $T_m$ | Minority | Female |
|---|---|---|---|---|---|
| II-COS | 0.16 | 5.56 | 2760 | 0.05 | 0.09 |
| SAST | 0.19 | 30.90 | 1200 | 0.03 | 0.05 |
| CP | 0.42 | 18.84 | 283 | 0.06 | 0.13 |

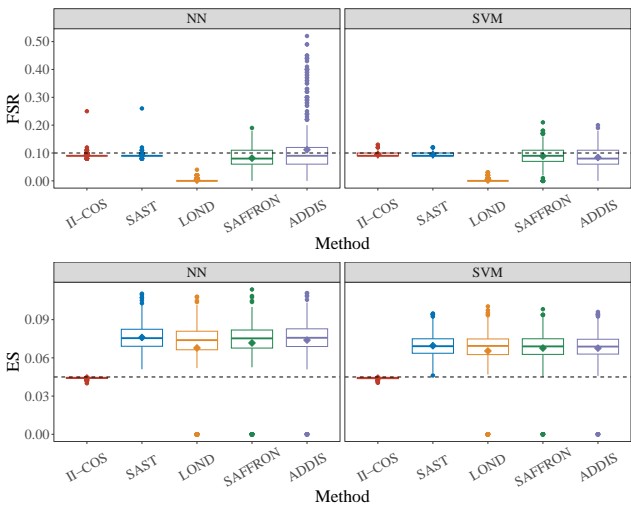

Figure 12: Boxplots of $\text{FSR}(T_m)$ and $\text{ES}(T_m)$ with two different learning algorithms under classification setting for II-COS, SAST, LOND, SAFFRON and ADDIS. The black dashed lines are the corresponding FSR level $\alpha = 0.1$ and the ES level $K = 0.045$.

## D.8 Experiments for the comparison with oracle procedure

Regarding efficiency, we conducted an experiment to compare the effectiveness of II-COS with an oracle method possessing knowledge of true state $\theta_t$. This can be succinctly formulated as follows. At time $t$, $\delta_t = 1$ if $\theta_t = 1$ and $\text{PC}(\boldsymbol{\delta}^{T_m})/\text{PS}(\boldsymbol{\delta}^{T_m}) \leq K$, the process halts when $\sum_{i \leq t} \delta_i = m$. To ensure a fair comparison, we fix $\alpha = 0.01$ for II-COS and choose $K = 0.045$ for both methods. The results are shown in Table 7. The FSR of the oracle procedure is 0 as expected. The $T_m$ of II-COS is very close to the oracle. This close proximity indicates the high efficiency of II-COS. Both FSR and ES are effectively controlled by II-COS.

## D.9 Experiments for the Extended II-COS under Varying Proportion Case

Let $N$ be a prespecified large integer denoting the number of samples arriving sequentially. Denote $n$ as the size of available labeled history dataset $\mathcal{D}$. We fix $N = 5000$, $n = 5000$. Consider three different varying patterns for $\pi_t$:

Table 7: $\text{FSR}(\boldsymbol{\delta}^{T_m})$, $\text{ES}(\boldsymbol{\delta}^{T_m})$ ($\times 10^{-2}$) and $T_m$ of the II-COS and the oracle procedure under classification model.

| Method | FSR | ES | $T_m$ |
|---|---|---|---|
| II-COS | 0.0097 | 4.41 | 686.245 |
| Oracle | 0 | 4.41 | 681.665 |

1. Blocks pattern: $\pi_t = 0.3$, for $t \in [1, 100] \cup [501, 600] \cup [1001, 1100] \cup [1501, 2000] \cup [2001, 2100] \cup [2501, 2600] \cup [3001, 3100] \cup [3501, 4000]$; $\pi_t = 0.2$ for $t \in [101, 500] \cup [601, 1000] \cup [1101, 1500] \cup [2101, 2500] \cup [2601, 3000] \cup [3101.3500] \cup [4000, 5000]$.

2. Linear pattern: Vary $\pi_t$ linearly from 0 to 0.5,
$\pi_t = 0.5t/N$.

3. Sine pattern: $\pi_t = \{\sin(8\pi t/N) + 1\}/4$, $\pi_t$ ranges between 0 and 0.5.

The other settings are the same as those in classification setting. For the historical dataset, we fix $\pi_t = 0.2$.

In simulation we generate 500 data points prior to $t = 1$ to form an initial proportion estimate. The varying proportion estimates are updated every 200 time points. The bandwidth parameter $b$ for estimating $\pi_t^\lambda$ are chosen based on normal reference rule. We set $b = 38$. The window size $q$ are chosen by a rule of thumb in practice. Let $q = Cb$, where $C$ is a fixed constant between 10 and 20. In our simulations, we fix $q = 500$ and $\lambda = 0.5$. As for the density estimation, we first estimate $f_0$ and $f_1$ on the calibration data, then compute $\widehat{f}^t = \widehat{f}_0(w)(1 - \widehat{\pi}_t) + \widehat{f}_1(w)\widehat{\pi}_t$. Thus accordingly, we estimate $L_t$ by $\widehat{L}_t^\lambda = \frac{(1-\widehat{\pi}_t^\lambda)\widehat{f}_0(W_t)}{\widehat{f}^t(W_t)} \wedge 1$. As for those methods based on conformal $p$-values, they do not need modification since they only require the exchangeability of data in the null hypothesis. The varying proportion case does not violate such a condition.

The simulation results are shown in Figures 13-15, from which we can observe that the extended version of II-COS performs better than the benchmarks under all three different varying settings for both $C_1$(FSR) and $C_2$(ES) control.

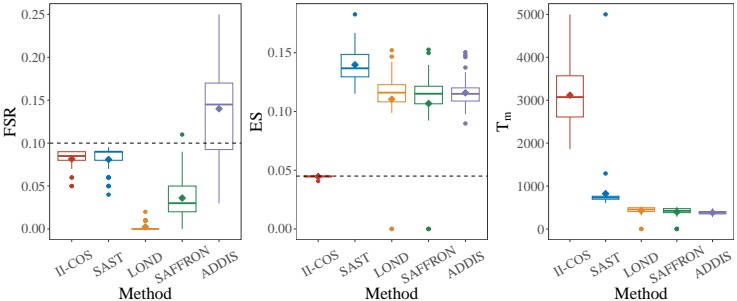

Figure 13: Boxplots of FSR($\boldsymbol{\delta}^{T_m}$), ES($\boldsymbol{\delta}^{T_m}$) and $T_m$ for II-COS, SAST, LOND, SAFFRON and ADDIS (Blocks pattern). The black dashed lines indicate the corresponding nominal levels.

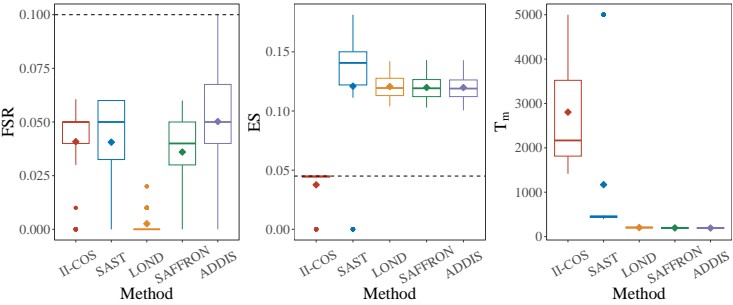

Figure 14: Boxplots of FSR($\boldsymbol{\delta}^{T_m}$), ES($\boldsymbol{\delta}^{T_m}$) and $T_m$ for II-COS, SAST, LOND, SAFFRON and ADDIS (Linear pattern). The black dashed lines indicate the corresponding nominal levels.

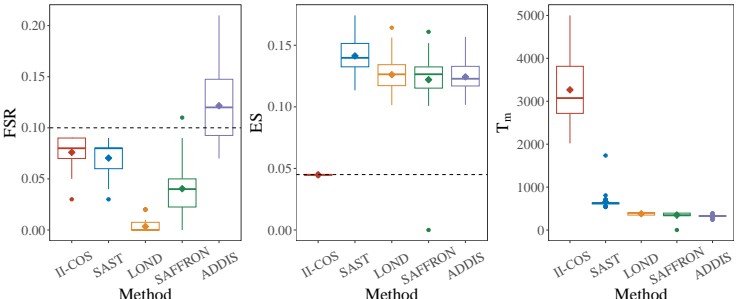

Figure 15: Boxplots of $\mathrm{FSR}(\boldsymbol{\delta}^{T_m})$, $\mathrm{ES}(\boldsymbol{\delta}^{T_m})$ and $T_m$ for II-COS, SAST, LOND, SAFFRON and ADDIS (Sine pattern). The black dashed lines indicate the corresponding nominal levels.

# E    Proofs

## E.1    Auxiliary Lemmas

The following results include a standard uniform bound for kernel density estimator [36, 20] and a simple corollary from the central limit theorem [11].

**Lemma E.1.** *If Assumption 3.1 hold and we take the bandwidth* $h = n_1^{-1/(2\beta+1)}$*, then with probability at least* $1 - 1/n_1$

$$\sup_{w \in \mathbb{R}} |\widehat{f}(w) - f(w)| \leq D_1 n_1^{\frac{-\beta}{2\beta+1}} \sqrt{\log n_1},$$

*where* $D_1 = D_1(M, c_\beta, \beta, K)$ *is a positive constant depending on* $M$ *and* $\beta, c_\beta$ *of Hölder continuity and the kernel* $K(\cdot)$.

**Lemma E.2.** *The estimation* $\widehat{\pi}$ *satisfies*

$$|\widehat{\pi} - \pi| \leq \sqrt{\pi(1-\pi)} n_1^{-\frac{1}{2}+\gamma},$$

*with probability* $1 - n_1^{-2\gamma}$ *for any constant* $0 < \gamma < 1/2$.

Next lemma characterizes the uniform convergence of $\widehat{L}(w)$.

**Lemma E.3** (Uniform convergence of $\widehat{L}(w)$). *Suppose Assumption 3.1 holds. Taking the bandwidth of kernel density estimator as* $h = n_1^{-1/(2\beta+1)}$*, then we have*

$$\sup_{w \in \mathbb{R}} |\widehat{L}(w) - L(w)| \leq D_2 n_1^{\frac{-\beta}{2\beta+1}} \sqrt{\log n_1},$$

*for some positive constant* $D_2 = D_2(M, D_1, \ell, \pi)$ *with probability* $1 - 2/n_1 - 1/n_1^{1/3}$.

*Proof.*

Note that $\widehat{L}(w) = (1 - \widehat{\pi})\frac{\widehat{f}_0(w)}{\widehat{f}(w)} \wedge 1$ and

$$\sup_{w \in \mathbb{R}} |\widehat{L}(w) - L(w)| \leq \sup_{w \in \mathbb{R}} \left| (1 - \widehat{\pi})\frac{\widehat{f}_0(w)}{\widehat{f}(w)} - L(w) \right| = \sup_{w \in \mathbb{R}} \left| \frac{(1 - \widehat{\pi})\widehat{f}_0(w)f(w) - (1 - \pi)f_0(w)\widehat{f}(w)}{\widehat{f}(w)f(w)} \right|$$

$$\leq \sup_{w \in \mathbb{R}} \frac{1}{\widehat{f}(w)f(w)} \left\{ (1 - \widehat{\pi})f(w)|\widehat{f}_0(w) - f_0(w)| + |\pi - \widehat{\pi}|f_0(w)f(w) + (1 - \pi)f_0(w)|\widehat{f}(w) - f(w)| \right\}$$

$$\leq \sup_{w \in \mathbb{R}} \frac{|\widehat{f}_0(w) - f_0(w)| + f_0(w)|\pi - \widehat{\pi}|}{\widehat{f}(w)} + \sup_{w \in \mathbb{R}} \frac{L(w)|\widehat{f}(w) - f(w)|}{\widehat{f}(w)}$$

$$\overset{(i)}{\leq} \frac{1}{\inf_{w \in \mathbb{R}} \widehat{f}(w)} \{ D_1 n_1^{\frac{-\beta}{2\beta+1}} \sqrt{\log n_1} + M\sqrt{\pi(1 - \pi)}n_1^{-\frac{1}{2}+\gamma} + D_1 \cdot n_1^{\frac{-\beta}{2\beta+1}} \sqrt{\log n_1} \}$$

$$\overset{(ii)}{\leq} \frac{2}{\ell} \{ D_1^* n_1^{\frac{-\beta}{2\beta+1}} \sqrt{\log n_1} + M\sqrt{p(1 - p)}n_1^{-\frac{1}{2}+\gamma} + D_1 \cdot n_1^{\frac{-\beta}{2\beta+1}} \sqrt{\log n_1} \}$$

$$\overset{(iii)}{\leq} D_2 n_1^{\frac{-\beta}{2\beta+1}} \sqrt{\log n_1},$$

where $D_1^* = D_1^*(D_1, \pi)$ is a positive constant depending on $D_1$ and $\pi$ because $\widehat{f}_0$ is estimated by about $n_1(1 - \pi)$ samples. The $(i)$ follows the results directly in Lemma E.1 and Lemma E.2 and the bounds of $f(w)$ and $f_0(w)$ in Assumption 3.1-(1) with probability $1 - 2/n_1 - 1/n_1^{2\gamma}$. The $(ii)$ holds since

$$\widehat{f}(w) \geq f(w) - D_1 n_1^{\frac{-\beta}{2\beta+1}} \sqrt{\log n_1} \geq \ell - D_1 n_1^{\frac{-\beta}{2\beta+1}} \sqrt{\log n_1} > \ell/2$$

for sufficiently large $n_1$. Due to the fact $\beta \leq 1$ and taking $\gamma = 1/6$ in Lemma E.2, the $(iii)$ holds with $D_2 = 2\{D_1^* + M\sqrt{\pi(1 - \pi)} + D_1\}/\ell$. Hence we verify the conclusion. $\qquad\square$

The next lemma shows the property of data-driven II-COS procedure which is useful to give the bound of the mES gap.

**Lemma E.4.** *Suppose $\delta^t$ is the decision result at time $t \geq T_s$ by the data-driven II-COS procedure with $\sum_{i \leq t} \widehat{L}_i \delta_i / 1 \vee \sum_{i \leq t} \delta_i \leq \alpha'$, where $T_s = \inf\{t : \sum_{i=1}^t \delta_i = s\}$ and $s \geq 2$. Then we have*

$$\frac{\mathbb{E}\left( \sum\sum_{1 \leq i < j \leq t} \delta_i \delta_j \right)}{\mathbb{E}\left\{ \sum\sum_{1 \leq i < j \leq t} \delta_i \delta_j (1 - \widehat{L}_i)(1 - \widehat{L}_j) \right\}} \leq \left( 1 - \frac{2s}{s-1}\alpha' \right)^{-1}.$$

*Proof.* It suffices to show

$$\sum\sum_{1 \leq i < j \leq t} \delta_i \delta_j (1 - \widehat{L}_i)(1 - \widehat{L}_j) \geq \left( 1 - \frac{2s}{s-1}\alpha' \right) \sum\sum_{1 \leq i < j \leq t} \delta_i \delta_j.$$

By the selection procedure, we have

$$\sum_{i=1}^t \delta_i(1 - \widehat{L}_i) \geq (1 - \alpha') \sum_{i=1}^t \delta_i. \tag{9}$$

Squaring both sides of (9) and making some decomposition, we have

$$\sum\sum_{1 \leq i < j \leq t} \delta_i \delta_j (1 - \widehat{L}_i)(1 - \widehat{L}_j) \geq (1 - \alpha')^2 \sum\sum_{1 \leq i < j \leq t} \delta_i \delta_j + \frac{1}{2}\sum_{i=1}^t \{(1 - \alpha')^2 - (1 - \widehat{L}_i)^2\}\delta_i.$$

Notice that

$$\sum_{i=1}^t \delta_i = \frac{2}{s-1} \sum\sum_{1 \leq i < j \leq t} \delta_i \delta_j.$$

We have

$$(1 - \alpha')^2 \sum\sum_{1 \leq i < j \leq t} \delta_i \delta_j + \frac{1}{2} \sum_{i=1}^{t} \{(1 - \alpha')^2 - (1 - \widehat{L}_i)^2\} \delta_i$$

$$\geq (1 - \alpha')^2 \sum\sum_{1 \leq i < j \leq t} \delta_i \delta_j + \frac{1}{2} \sum_{i=1}^{t} \{(1 - \alpha')^2 - 1\} \delta_i$$

$$\geq (1 - \alpha')^2 \sum\sum_{1 \leq i < j \leq t} \delta_i \delta_j - \frac{\alpha'(2 - \alpha')}{2} \frac{2}{s-1} \sum\sum_{1 \leq i < j \leq t} \delta_i \delta_j$$

$$\geq (1 - \frac{2s}{s-1} \alpha') \sum\sum_{1 \leq i < j \leq t} \delta_i \delta_j.$$

□

The next two lemmas will be used in the analysis of the extended II-COS for varying proportion case.

**Lemma E.5.** *Assume that random variables $Y_1$, $Y_2$, have density functions bounded by a constant $c > 0$, and other two random variables $Y_3$, and $Y_4$ satisfy $\mathbb{P}(|Y_1 - Y_3| \leq \varepsilon) \geq 1 - \zeta$ and $\mathbb{P}(|Y_2 - Y_4| \leq \epsilon) \geq 1 - \varsigma$, where $\varepsilon > 0$, $\epsilon > 0$, $\zeta > 0$ and $\varsigma > 0$. Then for all $t > 0$,*

$$|\mathbb{P}(Y_1 > t, Y_2 > t) - \mathbb{P}(Y_3 > t, Y_4 > t)| \leq 2c(\varepsilon + \epsilon) + 3(\zeta + \varsigma).$$

*Proof.* Define the events $\mathcal{E}_1 = \{|Y_1 - Y_3| \leq \varepsilon\}$, $\mathcal{E}_2 = \{|Y_2 - Y_4| \leq \epsilon\}$, and $\mathcal{E} = \mathcal{E}_1 \bigcup \mathcal{E}_2$. Note that

$$\mathbb{P}(Y_1 + \varepsilon > t, \mathcal{E}_1) \leq \mathbb{P}(Y_3 > t, \mathcal{E}_1) \leq \mathbb{P}(Y_1 - \varepsilon > t, \mathcal{E}_1),$$

and

$$\mathbb{P}(Y_2 + \epsilon > t, \mathcal{E}_2) \leq \mathbb{P}(Y_4 > t, \mathcal{E}_2) \leq \mathbb{P}(Y_2 - \epsilon > t, \mathcal{E}_2),$$

we have

$$\mathbb{P}(Y_1 > t, Y_2 > t) - \mathbb{P}(Y_3 > t, Y_4 > t)$$
$$\leq |\mathbb{P}(Y_1 > t, Y_2 > t) - \mathbb{P}(Y_3 > t, Y_4 > t, \mathcal{E})| + \mathbb{P}(\mathcal{E}_1^c) + \mathbb{P}(\mathcal{E}_2^c)$$
$$\leq |\mathbb{P}(Y_1 > t, Y_2 > t) - \mathbb{P}(Y_1 - \varepsilon > t, Y_2 - \epsilon > t, \mathcal{E})|$$
$$\quad + |\mathbb{P}(Y_1 > t, Y_2 > t) - \mathbb{P}(Y_1 + \varepsilon > t, Y_2 + \epsilon > t, \mathcal{E})| + \mathbb{P}(\mathcal{E}_1^c) + \mathbb{P}(\mathcal{E}_2^c)$$
$$\leq |\mathbb{P}(Y_1 > t, Y_2 > t) - \mathbb{P}(Y_1 - \varepsilon > t, Y_2 - \epsilon > t)| + 3\mathbb{P}(\mathcal{E}_1^c) + 3\mathbb{P}(\mathcal{E}_2^c)$$
$$\quad + |\mathbb{P}(Y_1 > t, Y_2 > t) - \mathbb{P}(Y_1 + \varepsilon > t, Y_2 + \epsilon > t)|$$
$$\leq 2c(\varepsilon + \epsilon) + 3(\zeta + \varsigma),$$

the last inequality holds because the density function is bounded.

□

**Lemma E.6.** *If $b^\zeta \leq q$ for some $\zeta > 1$ and $b$ is sufficiently large, the exponential weight $\kappa_b(s) = \exp\{-|s|/b\}$ satisfies*

$$\frac{\sum_{s=1}^{q} \kappa_b^2(s)}{\{\sum_{s=1}^{q} \kappa_b(s)\}^2} \leq C_b \frac{1}{b}$$

*and*

$$\frac{\sum_{s=1}^{q} s \kappa_b(s)}{\sum_{s=1}^{q} \kappa_b(s)} \leq C_b b,$$

*where $C_b > 0$ is constant determined by $b$.*

*Proof.* Take $Z_b = \exp\{-1/b\}$. Notice that

$$\sum_{s=i}^{j} \kappa_b(s) = \frac{\exp\{-(i+1)/b\} - \exp\{-(j+1)/b\}}{1 - \exp\{-1/b\}} = \frac{Z_b^{i+1} - Z_b^{j+1}}{1 - Z_b}.$$

Thus for a constant $C_b > 0$. we have

$$\frac{\sum_{s=1}^{q} \kappa_b^2(s)}{\{\sum_{s=1}^{q} \kappa_b(s)\}^2} = \left(\frac{Z_b^4 - Z_b^{2q}}{1 - Z_b^2}\right) \Big/ \left(\frac{Z_b^4 - 2Z_b^{q+2} + Z_b^{2q}}{1 + Z_b^2 - 2Z_b}\right)$$

$$\overset{(i)}{\leq} C_b(1 - Z_b)$$

$$\overset{(ii)}{\leq} C_b \frac{1}{b},$$

where $(i)$ holds since $b$ is sufficiently large such that $Z_b^q = \exp\{-q/b\} \leq \exp\{-b^{\zeta-1}\}$ can be eliminated and $Z_b < 1$. The last inequality $(ii)$ holds by $\exp\{x\} \geq x + 1$.

By the same discussion, we have

$$\frac{\sum_{s=1}^{q} s\kappa_b(s)}{\sum_{s=1}^{q} \kappa_b(s)} \leq \int_1^\infty sZ_b^s ds \Big/ \left(\frac{Z_b^2 - Z_b^q}{1 - Z_b}\right)$$

$$\leq \left(\frac{Z_b}{(1 - Z_b)^2}\right) \Big/ \left(\frac{Z_b^2 - Z_b^q}{1 - Z_b}\right)$$

$$\leq C_b \frac{1}{1 - Z_b}$$

$$\leq C_b(b + 1).$$

The last inequality holds since $\exp\{x\} \leq 1/(1 - x)$ for $x < 1$. For simplicity, we use the same notation $C_b$ to denote the constants. $\square$

## E.2 Proof of Proposition 2.1

*Proof.* For the part of individual constraint control, note that $L_i$ is defined as

$$L_i = \Pr(\theta_i = 0 \mid W_i) = \mathbb{E}\left[(1 - \theta_i) \mid W_i\right].$$

Define $\mathcal{W}^* = \sigma(W_1, \cdots)$. The stopping time $T$ is measurable respect to $\mathcal{W}^*$. The individual constraint at time $T$ satisfies

$$
\begin{aligned}
C_1(\boldsymbol{\delta}^T) &= \mathbb{E}\left\{ \frac{\sum_{i \leq T}\{(1 - \theta_i)G_0(\mathbf{X}_i) + \theta_i G_1(\mathbf{X}_i)\}\delta_i}{(\sum_{i \leq T} \delta_i) \vee 1} \right\} \\
&= \mathbb{E}\left[ \mathbb{E}\left\{ (R_T \vee 1)^{-1} \sum_{i \leq t}\{(1 - \theta_i)G_0(\mathbf{X}_i) + \theta_i G_1(\mathbf{X}_i)\}\delta_i \mid \mathcal{W}^* \right\} \right] \\
&\overset{(i)}{=} \mathbb{E}\left[ (R_t \vee 1)^{-1} \sum_{i \leq t}\{\mathbb{E}\{(1 - \theta_i) \mid W_i\}G_0(\mathbf{X}_i) + \mathbb{E}\{\theta_i \mid W_i\}G_1(\mathbf{X}_i)\}\delta_i \right] \\
&= \mathbb{E}\left\{ (R_t \vee 1)^{-1} \sum_{i \leq t}\{L_i G_0(\mathbf{X}_i) + (1 - L_i)G_1(\mathbf{X}_i)\}\delta_i \right\}.
\end{aligned}
$$

The $(i)$ holds since $\mathbf{X}_i$'s are independent of each other. By construction of the selection rule, we have that at the stopping time $T$, $((R_T \vee 1)^{-1} \sum_{i \leq T}\{L_i W_0(\mathbf{X}_i) + (1 - L_i)W_1(\mathbf{X}_i)\}\delta_i \leq \alpha$. It follows that $C_1(\boldsymbol{\delta}^T) \leq \alpha$ at a random time $T$. By construction of the selection rule, we have for stopping time $T$,

$$\sum_{1 \leq i < j \leq T} \sum g(\mathbf{X}_i, \mathbf{X}_j)\delta_i\delta_j(1 - L_i)(1 - L_j) \leq K \times \sum_{1 \leq i < j \leq T} \sum \delta_i\delta_j(1 - L_i)(1 - L_j). \quad (10)$$

Taking expectations on both sides of inequality (10) and by double expectation theorem, we finally obtain

$$\mathbb{E}\left\{ \sum_{1 \leq i < j \leq T} \sum g(\mathbf{X}_i, \mathbf{X}_j)\delta_i\delta_j\theta_i\theta_j \right\} \leq K \times \mathbb{E}\left\{ \sum_{1 \leq i < j \leq T} \sum \delta_i\delta_j\theta_i\theta_j \right\}.$$

It follows that for every time $t \geq T_2$,

$$C_2(\boldsymbol{\delta}^T) = \frac{\mathbb{E}\Big\{\sum\sum_{1 \leq i < j \leq T} g(\mathbf{X}_i, \mathbf{X}_j)\delta_i\delta_j\theta_i\theta_j\Big\}}{\mathbb{E}\Big\{(\sum_{i \leq T}\delta_i\theta_i)(\sum_{i \leq T}\delta_i\theta_i - 1)\Big\}} \leq K$$

$\square$

## E.3  Proof of Theorem 3.3

*Proof.* In the data-driven II-COS procedure, $\delta_i$ is determined by the estimated lFDR $\widehat{L}_i$ and we have

$$\frac{1}{R_t}\sum_{i \leq t}\{\widehat{L}_i G_0(\mathbf{X}_i) + (1 - \widehat{L}_i)G_1(\mathbf{X}_i)\}\delta_i \leq \alpha.$$

Note that

$$C_1(\boldsymbol{\delta}^t) = \mathbb{E}\Big\{\frac{1}{R_t}\sum_{i \leq t}\{L_i G_0(\mathbf{X}_i) + (1 - L_i)G_1(\mathbf{X}_i)\}\delta_i\Big\}$$

$$= \mathbb{E}\Big\{\frac{1}{R_t}\sum_{i \leq t}\{\widehat{L}_i G_0(\mathbf{X}_i) + (1 - \widehat{L}_i)G_1(\mathbf{X}_i)\}\delta_i\Big\} + \mathbb{E}\Big\{\frac{1}{R_t}\sum_{i \leq t}\{(L_i - \widehat{L}_i)G_0(\mathbf{X}_i) + (\widehat{L}_i - L_i)G_1(\mathbf{X}_i)\}\delta_i\Big\}$$

$$\leq \alpha + \mathbb{E}\Big\{\frac{1}{R_t}\sum_{i \leq t}\{(L_i - \widehat{L}_i)G_0(\mathbf{X}_i) + (\widehat{L}_i - L_i)G_1(\mathbf{X}_i)\}\delta_i\Big\}.$$

It suffices to bound the absolute value of the second term. We have

$$\Big|\mathbb{E}\Big\{\frac{1}{R_t}\sum_{i \leq t}\{(L_i - \widehat{L}_i)G_0(\mathbf{X}_i) + (\widehat{L}_i - L_i)G_1(\mathbf{X}_i)\}\delta_i\Big\}\Big|$$

$$\leq \mathbb{E}\Big\{\frac{1}{R_t}\sum_{i \leq t}\{|L_i - \widehat{L}_i||G_0(\mathbf{X}_i)| + |\widehat{L}_i - L_i||G_1(\mathbf{X}_i)|\}\delta_i\Big\}$$

$$\leq \mathbb{E}\Big\{\frac{1}{R_t}\sum_{i \leq t}\{\sup_{w \in \mathcal{R}}|L_i - \widehat{L}_i||G_0(\mathbf{X}_i)| + \sup_{w \in \mathcal{R}}|\widehat{L}_i - L_i||G_1(\mathbf{X}_i)|\}\delta_i\Big\}$$

$$\leq 2c_G D_2 n_1^{\frac{-\beta}{2\beta+1}}\sqrt{\log n_1} + 2n_1^{-1} + n_1^{-\frac{1}{3}}.$$

The last inequality follows from Lemma E.3 and $\max_w |\widehat{L}(w) - L(w)| \leq 1$. Notice that $n_1^{-\frac{1}{3}} \leq n_1^{\frac{-\beta}{2\beta+1}}$ by $\beta \leq 1$. We have

$$C_1(\boldsymbol{\delta}^t) \leq \alpha + D n_1^{\frac{-\beta}{2\beta+1}}\sqrt{\log n_1},$$

where $D = 2c_G D_2 + 3.$ $\square$

## E.4  Proof of Theorem 3.4

*Proof.* Notice that

$$\mathbb{E}\Big\{\Big|\sum\sum_{1 \leq i < j \leq t} g(\mathbf{X}_i, \mathbf{X}_j)\delta_i\delta_j(1 - \widehat{L}_i)(1 - \widehat{L}_j) - \sum\sum_{1 \leq i < j \leq t} g(\mathbf{X}_i, \mathbf{X}_j)\delta_i\delta_j(1 - L_i)(1 - L_j)\Big|\Big\}$$

$$\leq \mathbb{E}\Big\{\sum\sum_{1 \leq i < j \leq t} g(\mathbf{X}_i, \mathbf{X}_j)\delta_i\delta_j(|\widehat{L}_i - L_i| + |\widehat{L}_j - L_j| + \widehat{L}_i|\widehat{L}_j - L_j| + L_j|\widehat{L}_i - L_i|)\Big\}$$

$$\overset{(i)}{\leq} 2c_g \mathbb{E}\Big\{\sum\sum_{1 \leq i < j \leq t} \delta_i\delta_j(|\widehat{L}_i - L_i| + |\widehat{L}_j - L_j|)\Big\}$$

$$\overset{(ii)}{\leq} 2c_g D n_1^{\frac{-\beta}{2\beta+1}}\sqrt{\log n_1}\,\mathbb{E}\Big(\sum\sum_{1 \leq i < j \leq t} \delta_i\delta_j\Big). \tag{11}$$

The $(i)$ holds by Assumption 3.2 and $L(w), \widehat{L}(w) \leq 1$ even when $t$ is random, as the uniform convergence of $\widehat{L}(w)$. And $(ii)$ follows from Lemma E.3. By the similar arguments we have

$$\left|\mathbb{E}\Big\{\sum_{1\leq i<j\leq t}\delta_i\delta_j(1-\widehat{L}_i)(1-\widehat{L}_j) - \sum_{1\leq i<j\leq t}\delta_i\delta_j(1-L_i)(1-L_j)\Big\}\right|$$

$$\leq 2Dn_1^{\frac{-\beta}{2\beta+1}}\sqrt{\log n_1}\mathbb{E}(\sum_{1\leq i<j\leq t}\delta_i\delta_j). \tag{12}$$

Denote $\Delta_{n_1} = Dn_1^{\frac{-\beta}{2\beta+1}}\sqrt{\log n_1}$. Combining (11) and (12), it follows that for every time $t \geq T_s$,

$$C_2(\boldsymbol{\delta}^t) = \frac{\mathbb{E}\Big\{\sum_{1\leq i<j\leq t} g(\mathbf{X}_i, \mathbf{X}_j)\delta_i\delta_j(1-L_i)(1-L_j)\Big\}}{\mathbb{E}\Big\{\sum_{1\leq i<j\leq t}\delta_i\delta_j(1-L_i)(1-L_j)\Big\}}$$

$$\overset{(i)}{\leq} \frac{\mathbb{E}\Big\{\sum_{1\leq i<j\leq t} g(\mathbf{X}_i, \mathbf{X}_j)\delta_i\delta_j(1-\widehat{L}_i)(1-\widehat{L}_j)\Big\} + 2c_g\Delta_{n_1}\mathbb{E}(\sum_{1\leq i<j\leq t}\delta_i\delta_j)}{\mathbb{E}\Big\{\sum_{1\leq i<j\leq t}\delta_i\delta_j(1-L_i)(1-L_j)\Big\}}$$

$$\overset{(ii)}{\leq} \Big[K + 2c_g\Delta_{n_1}\frac{\mathbb{E}(\sum_{1\leq i<j\leq t}\delta_i\delta_j)}{\mathbb{E}\Big\{\sum_{1\leq i<j\leq t}\delta_i\delta_j(1-\widehat{L}_i)(1-\widehat{L}_j)\Big\}}\Big] \times \frac{\mathbb{E}\Big\{\sum_{1\leq i<j\leq t}\delta_i\delta_j(1-\widehat{L}_i)(1-\widehat{L}_j)\Big\}}{\mathbb{E}\Big\{\sum_{1\leq i<j\leq t}\delta_i\delta_j(1-L_i)(1-L_j)\Big\}}$$

$$\overset{(iii)}{\leq} \Big\{K + \frac{2c_g\Delta_{n_1}}{1-\frac{2m\alpha'}{m-1}}\Big\} \times \frac{\mathbb{E}\Big\{\sum_{1\leq i<j\leq t}\delta_i\delta_j(1-\widehat{L}_i)(1-\widehat{L}_j)\Big\}}{\mathbb{E}\Big\{\sum_{1\leq i<j\leq t}\delta_i\delta_j(1-L_i)(1-L_j)\Big\}}$$

$$\overset{(iv)}{\leq} \Big\{K + \frac{2c_g\Delta_{n_1}}{1-\frac{2m\alpha'}{m-1}}\Big\} \times \Big[1 - 2\Delta_{n_1}\frac{\mathbb{E}(\sum_{1\leq i<j\leq t}\delta_i\delta_j)}{\mathbb{E}\Big\{\sum_{1\leq i<j\leq t}\delta_i\delta_j(1-\widehat{L}_i)(1-\widehat{L}_j)\Big\}}\Big]^{-1}$$

$$\overset{(v)}{\leq} \Big\{K + \frac{2c_g\Delta_{n_1}}{1-\frac{2m\alpha'}{m-1}}\Big\}\Big\{1 - \frac{2\Delta_{n_1}}{1-\frac{2m\alpha'}{m-1}}\Big\}^{-1}.$$

The $(i)$ follows from (11), and $(ii)$ comes from the operation of our algorithm, where

$$\mathbb{E}\Big\{\sum_{1\leq i<j\leq t} g(\mathbf{X}_i, \mathbf{X}_j)\delta_i\delta_j(1-\widehat{L}_i)(1-\widehat{L}_j)\Big\} \leq K \cdot \mathbb{E}\Big\{\sum_{1\leq i<j\leq t}\delta_i\delta_j(1-\widehat{L}_i)(1-\widehat{L}_j)\Big\}.$$

The $(iii)$ and $(v)$ are directly from Lemma E.4. The $(iv)$ holds due to (12). Thus we have

$$C_2(\boldsymbol{\delta}^t) \leq K + \frac{(K+c_g)\Delta_{n_1}}{0.5 - \frac{m\alpha'}{m-1} - \Delta_{n_1}}$$

$\square$

## E.5  Proof of Corollary 3.5

*Proof.* Define the $C_1$ constraint conditional on $\mathcal{W}$ as

$$C_1'(\boldsymbol{\delta}^t) = \frac{1}{R_t}\sum_{i\leq t}\{(L_i - \widehat{L}_i)G_0(\mathbf{X}_i) + (\widehat{L}_i - L_i)G_1(\mathbf{X}_i)\}\delta_i.$$

By the proofs of Theorem 3.3, we have for any $t$

$$C_1'(\boldsymbol{\delta}^t) \leq \alpha + 2c_G \sup_{w\in\mathcal{R}}|\widehat{L}(w) - L(w)|.$$

Hence at stopping time $T_m$, we have

$$C_1'(\boldsymbol{\delta}^{T_m}) = \mathbb{E}[C_1'(\boldsymbol{\delta}^{T_m})] \leq \mathbb{E}[\sup_t C_1'(\boldsymbol{\delta}^t)] \leq \alpha + 2c_G \mathbb{E}[\sup_{w \in \mathcal{R}} |\widehat{L}(w) - L(w)|] \leq \alpha + \Delta_{n_1}.$$

By the proof of Theorem 3.4, we can also drop off the expectation and replace $\Delta_{n_1}$ with $\sup_{w \in \mathcal{R}} |\widehat{L}(w) - L(w)|$. That is

$$\left\{1 - \frac{2 \sup_{w \in \mathcal{R}} |\widehat{L}(w) - L(w)|}{1 - \frac{2m\alpha}{m-1}}\right\} \sum_{1 \leq i < j \leq t} \sum g(\mathbf{X}_i, \mathbf{X}_j) \delta_i \delta_j (1 - L_i)(1 - L_j)$$

$$\leq \left\{K + \frac{2c_g \sup_{w \in \mathcal{R}} |\widehat{L}(w) - L(w)|}{1 - \frac{2m\alpha}{m-1}}\right\} \sum_{1 \leq i < j \leq t} \sum \delta_i \delta_j (1 - L_i)(1 - L_j). \tag{13}$$

Hence at stopping time $t = T_m$, (13) still holds. Take expectations at both sides and we will get

$$C_2(\boldsymbol{\delta}^{T_m}) \leq K + \frac{(K + c_g)\Delta_{n_1}}{0.5 - \frac{m\alpha'}{m-1} - \Delta_{n_1}}.$$

Notice that $\Delta_{n_1} = D n_1^{\frac{-\beta}{2\beta+1}} \sqrt{\log n_1}$ converges to 0 as $n_1 \to \infty$ and $\alpha' < (1 - 1/m)/2$ by the condition. It follows that

$$\lim_{n_1 \to \infty} C_1(\boldsymbol{\delta}^{T_m}) \leq \alpha \quad \text{and} \quad \lim_{n_1 \to \infty} C_2(\boldsymbol{\delta}^{T_m}) \leq K.$$

### E.6 Proof of Theorem C.1

*Proof.* We first denote some notations. For notational simplicity, we use $c$ to denote constants in the following proofs.

Recall the conformal score function $Q(\mathbf{X}_t)$. The conformal $p$-value is

$$\widehat{p}_t = \frac{1 + \sum_{i \in \mathcal{L}} \mathbb{I}\{Q(\tilde{W}_i) \leq Q(W_t)\}}{1 + |\mathcal{L}|} = \widehat{F}_{n_2}(Z_t),$$

where $n_2 = |\mathcal{L}|$.

Denote the population $p$-value as $p_t = F(Q(W_t))$, where $F(\cdot)$ is the distribution of $Q(W_t)$ conditional on $\theta_t = 0$. Denote $\pi_t = \Pr(\theta_t = 1)$ and

$$\pi_t^\lambda = 1 - \frac{\Pr(p_t > \lambda)}{1 - \lambda}.$$

The corresponding local FDR is defined as

$$L_t^\lambda = \frac{\pi_t^\lambda f_0(W_t)}{f^t(W_t)}.$$

It follows the definition of $C_1^\lambda(\boldsymbol{\delta}^t)$ and $C_2^\lambda(\boldsymbol{\delta}^t)$.

$$C_1^\lambda(\boldsymbol{\delta}^t) = \mathbb{E}\left\{\frac{1}{R_t} \sum_{i \leq t} \{L_i^\lambda G_0(\mathbf{X}_i) + (1 - L_i^\lambda)G_1(\mathbf{X}_i)\}\delta_i\right\}$$

and

$$C_2^\lambda(\boldsymbol{\delta}^t) = \frac{\mathbb{E}\left\{\sum_{1 \leq i < j \leq t} \sum g(\mathbf{X}_i, \mathbf{X}_j)\delta_i\delta_j(1 - L_i^\lambda)(1 - L_j^\lambda)\right\}}{\mathbb{E}\left\{\sum_{1 \leq i < j \leq t} \sum g(\mathbf{X}_i, \mathbf{X}_j)\delta_i\delta_j\right\}}.$$

The estimated proportion is defined as

$$\widehat{\pi}_t^\lambda = 1 - \frac{\sum_{j=t-q}^{t-1} \kappa_b(j-t)\mathbb{I}\{\widehat{p}_j > \lambda\}}{(1 - \lambda)\sum_{j=t-q}^{t-1} \kappa_b(j-t)}.$$

And denote the ideal estimated proportion via population p-values as

$$\tilde{\pi}_t^\lambda = 1 - \frac{\sum_{j=t-q}^{t-1} \kappa_b(j-t)\mathbb{I}\{p_j > \lambda\}}{(1-\lambda)\sum_{j=t-q}^{t-1} \kappa_b(j-t)},$$

where $\kappa_b(s) = \exp\{-|s|/b\}$ and $q$ is the size of a neighborhood for estimation before time $t$.

We first claim the following proposition and the proof is deferred in Section E.7.

**Proposition E.7.** *Suppose the assumptions in Theorem 3 hold. Then*

$$|\widehat{\pi}^\lambda - \pi^\lambda| \le c \max\left\{\frac{1}{b}, \frac{\sqrt{\log n_1}}{n_1^{1/2}}, (b\eta)^2\right\}^{1/2-\gamma} \tag{14}$$

*with probability $1 - \max\left\{1/b, \sqrt{\log n_1/n_1}, (b\eta)^2\right\}^{2\gamma}$, where $c > 0$ is a constant and $0 < \gamma < 1/2$.*

Notice that

$$\sup_w |\widehat{f}^t(w) - f^t(w)|$$
$$= \sup_w |\widehat{f}_0(w)(1-\widehat{\pi}_t) + \widehat{f}_1(w)\widehat{\pi}_t - f_0(w)(1-\pi_t) - f_1(w)\pi_t|$$
$$\le \sup_w |\widehat{f}_0(w) - f_0(w)| + \sup_w |\widehat{f}_1(w) - f_1(w)| + 2M|\widehat{\pi}_t - \pi_t|.$$

The last inequality holds since $\widehat{\pi}_t \in [0,1]$ and $f_0(w)$ and $f_1(w)$ are upper bounded by $M$.

Thus by Lemma E.1 and take $\gamma$ in Proposition E.7 at $1/6$, with probability $1 - c\max\left\{1/b, \sqrt{\log n_1/n_1}, (b\eta)^2\right\}^{1/3}$ we have

$$\sup_w |\widehat{f}^t(w) - f^t(w)| \le cn_1^{\frac{-\beta}{2\beta+1}}\sqrt{\log n_1} + c\max\left\{b^{-\frac{1}{3}}, \left(\frac{\log n_1}{n_1}\right)^{\frac{1}{6}}, (b\eta)^{\frac{2}{3}}\right\}.$$

By the procedure of Lemma E.3 directly, with probability $1 - c\max\left\{1/b, \sqrt{\log n_1/n_1}, (b\eta)^2\right\}^{1/3}$ we have

$$\sup_w |\widehat{L}_t^\lambda(w) - L_t^\lambda(w)| \le cn_1^{\frac{-\beta}{2\beta+1}}\sqrt{\log n_1} + c\max\left\{b^{-\frac{1}{3}}, \left(\frac{\log n_1}{n_1}\right)^{\frac{1}{6}}, (b\eta)^{\frac{2}{3}}\right\}.$$

It follows that

$$C_1^\lambda(\boldsymbol{\delta}^t) \le \alpha + cn_1^{\frac{-\beta}{2\beta+1}}\sqrt{\log n_1} + 2cc_G\max\left\{b^{-\frac{1}{3}}, \left(\frac{\log n_1}{n_1}\right)^{\frac{1}{6}}, (b\eta)^{\frac{2}{3}}\right\}.$$

Denote $\Delta_{n_1}' = 2cc_G n_1^{\frac{-\beta}{2\beta+1}}\sqrt{\log n_1} + c\max\left\{b^{-\frac{1}{3}}, \left(\frac{\log n_1}{n_1}\right)^{\frac{1}{6}}, (b\eta)^{\frac{2}{3}}\right\}$. By our assumption, $b\eta = o(1)$. So

$$\lim_{b,n_1\to\infty} \Delta_{n_1}' = 0.$$

By the additional assumption that $\Pr(p_t > \lambda \mid \theta_t = 1) = 0$, we have $\pi_t^\lambda = \pi_t$. Thus Hence the first part of Theorem 3 is completed.

At last, we can use the same procedure to prove that

$$C_2^\lambda(\boldsymbol{\delta}^t) \le K + \frac{(K + c_g)\Delta_{n_1}'}{0.5 - \frac{m\alpha'}{m-1} - \Delta_{n_1}'}.$$

Since $\pi_t^\lambda = \pi_t$, $C_2^\lambda(\boldsymbol{\delta}^t) = C_2(\boldsymbol{\delta}^t)$ and the results directly follow.

## E.7 Proof of Proposition E.7

*Proof.* Notice that

$$\Pr(|\widehat{\pi}_t^\lambda - \pi_t^\lambda| > \varepsilon) \le \Pr\left(|\widehat{\pi}_t^\lambda - \tilde{\pi}_t^\lambda| > \frac{\varepsilon}{2}\right) + \Pr\left(|\tilde{\pi}_t^\lambda - \pi_t^\lambda| > \frac{\varepsilon}{2}\right).$$

We discuss for the two parts respectively. For the first term of the above inequality, it suffices to show the upper bound of the second moment of $|\widehat{\pi}_t^\lambda - \tilde{\pi}_t^\lambda|$.

For convenience, denote $U_j = \mathbb{I}(p_j > \lambda)$, and $V_j = \mathbb{I}(\widehat{p}_j > \lambda)$. It follows

$$\mathbb{E}\{|\widehat{\pi}_t^\lambda - \tilde{\pi}_t^\lambda|^2\}$$

$$= \frac{1}{\{\sum_{j=t-q}^{t-1} \kappa_b(j-t)\}^2(1-\lambda)^2} \mathbb{E}\left[\sum_{j=t-q}^{t-1} \kappa_b(j-t)(U_j - V_j)\right]^2$$

$$= \frac{\sum_{j=t-q}^{t-1} \kappa_b^2(j-t)\mathbb{E}\{(U_j - V_j)^2\}}{\{\sum_{j=t-q}^{t-1} \kappa_b(j-t)\}^2(1-\lambda)^2} + \frac{\sum_{i,j\in\mathcal{N}_q(t),q\ne j} \kappa_b(i-t)\kappa_b(j-t)\mathbb{E}\{(U_i - V_i)(U_j - V_j)\}}{\{\sum_{j=t-q}^{t-1} \kappa_b(j-t)\}^2(1-\lambda)^2}$$

$$= \frac{1}{\{\sum_{j=t-q}^{t-1} \kappa_b(j-t)\}^2(1-\lambda)^2}\left[\sum_{j=t-q}^{t-1} \kappa_b^2(j-t)\Big\{\Pr(p_j > \lambda) + \Pr(\widehat{p}_j > \lambda) - 2\Pr(p_j > \lambda, \widehat{p}_j > \lambda)\Big\}\right.$$

$$\left. +2\sum_{t-q\le i<j\le t-1} \kappa_b(i-t)\kappa_b(j-t)\Big\{\Pr(p_i > \lambda, p_j > \lambda) - 2\Pr(\widehat{p}_i > \lambda, p_j > \lambda) + \Pr(\widehat{p}_i > \lambda, \widehat{p}_j > \lambda)\Big\}\right].$$

Now we check the upper bound of $|p_j - \widehat{p}_j|$. We can rewrite them as $p_j = F(Q(W_j))$ and $\widehat{p}_j = \widehat{F}_{n_2}(Q(W_j))$. Even though $n_2$ is a random variable, due to the two-group model, $\widehat{F}_{n_2}(\cdot)$ is still an empirical distribution function composed by i.i.d. samples conditional on $\{\tilde{\theta}_i\}_{i\in\mathcal{C}}$ where $\mathcal{C}$ is the index set of calibration set. Thus by DKW inequality, for any $\varepsilon_1 > 0$

$$\Pr(|p_j - \widehat{p}_j| \le \varepsilon_1) = \mathbb{E}\left[\Pr\left(|p_j - \widehat{p}_j| \le \varepsilon_1 \mid \{\tilde{\theta}_i\}_{i\in\mathcal{C}}\right)\right]$$

$$\le \mathbb{E}\left[\Pr\left(\sup_z \left|F(z) - \widehat{F}_{n_2}(z)\right| \le \varepsilon_1 \mid \{\tilde{\theta}_i\}_{i\in\mathcal{C}}\right)\right]$$

$$\le \mathbb{E}\left[1 - 2\exp\{-n_2\varepsilon_1^2\}\right]$$

$$\le 1 - 2\exp\{-(1-\pi)n_1\varepsilon_1^2\} - 4\pi/\{n_1(1-\pi)\}.$$

The last inequality holds since $n_2 \ge (1-\pi)n_1/2$ with probability $1 - 4\pi/\{n_1(1-\pi)\}$.

Thus by Lemma E.5 and the fact that $p_j$ has a bounded density function, we obtain

$$|\Pr(p_j > \lambda) - \Pr(\widehat{p}_j > \lambda)| \le 2\varepsilon_1 + 6\exp\{-(1-\pi)n_1\varepsilon_1^2\} + 12\pi/\{n_1(1-\pi)\}.$$

Take $\varepsilon_1 = \sqrt{\log n_1/\{n_1(1-\pi)\}}$, for sufficient large $n_1$ such that $\sqrt{n_1 \log n_1} \ge 6 + 6\pi$ we have

$$|\Pr(p_j > \lambda) - \Pr(\widehat{p}_j > \lambda)| \le \frac{2\sqrt{\log n_1}}{(1-\pi)n^{1/2}} + \frac{6 + 12\pi/(1-\pi)}{n_1} \le \frac{3\sqrt{\log n_1}}{(1-\pi)n^{1/2}}.$$

Again by Lemma E.5 and the same $\varepsilon_1$, for any $i, j \in \{t-q, \ldots, t-1\} := \mathcal{N}_q(t)$, we have

$$|\Pr(p_j > \lambda, p_i > \lambda) - \Pr(\widehat{p}_j > \lambda, \widehat{p}_i > \lambda)| \le 4\varepsilon_1 + 12\exp\{-(1-\pi)n_1\varepsilon_1^2\} + \frac{24\pi}{n_1(1-\pi)} \le \frac{6\sqrt{\log n_1}}{(1-\pi)n_1^{1/2}}.$$

Therefore by Lemma E.6, we have

$$\mathbb{E}(|\widehat{\pi}_t^\lambda - \tilde{\pi}_t^\lambda|) \le \frac{9C_b\sqrt{\log n_1}}{bn_1^{1/2}(1-\pi)(1-\lambda)^2} + \frac{12\sqrt{\log n_1}}{n_1^{1/2}(1-\pi)(1-\lambda)^2}.$$

So by definition and Markov' inequality, we have

$$
\begin{aligned}
\Pr\left(|\widehat{\pi}_t^\lambda - \tilde{\pi}_t^\lambda| > \frac{\varepsilon}{2}\right) &\leq \frac{4\mathbb{E}(|\widehat{\pi}_t^\lambda - \tilde{\pi}_t^\lambda|)}{\varepsilon^2} \\
&\leq \frac{36 C_b \sqrt{\log n_1}}{b n_1^{1/2}(1-\pi)(1-\lambda)^2\varepsilon^2} + \frac{48\sqrt{\log n_1}}{n_1^{1/2}(1-\pi)(1-\lambda)^2\varepsilon^2}
\end{aligned}
$$

For the second part, we have

$$
\begin{aligned}
\Pr\left(|\tilde{\pi}_t^\lambda - \pi_t^\lambda| > \frac{\varepsilon}{2}\right) &= \Pr\left(\left|\frac{\Pr(p_t > \lambda)}{1-\lambda} - \frac{\sum_{j=t-q}^{t-1}\kappa_b(j-t)\mathbb{I}\{p_j > \lambda\}}{\{\sum_{j=t-q}^{t-1}\kappa_b(j-t)\}^2(1-\lambda)}\right| > \frac{\varepsilon}{2}\right) \\
&\leq \frac{4\mathbb{E}\left[\sum_{j=t-q}^{t-1}\kappa_b(j-t)\{\Pr(p_t > \lambda) - \mathbb{I}(p_j > \lambda)\}\right]^2}{\{\sum_{j=t-q}^{t-1}\kappa_b(j-t)\}^2\varepsilon^2(1-\lambda)^2} \\
&= \frac{4\sum_{j=t-q}^{t-1}\kappa_b^2(j-t)\left[\mathrm{Var}\{\mathbb{I}(p_j > \lambda)\} + \{\Pr(p_j > \lambda) - \Pr(p_t > \lambda)\}^2\right]}{\{\sum_{j=t-q}^{t-1}\kappa_b(j-t)\}^2\varepsilon^2(1-\lambda)^2} \\
&\overset{(i)}{\leq} \frac{C_b}{b\varepsilon^2(1-\lambda)^2} + \frac{4\sum_{j=t-q}^{t-1}\kappa_b^2(j-t)\{\Pr(p_j > \lambda) - \Pr(p_t > \lambda)\}^2}{\{\sum_{j=t-q}^{t-1}\kappa_b^2(j-t)\}^2\varepsilon^2(1-\lambda)^2} \\
&\overset{(ii)}{\leq} \frac{C_b}{b\varepsilon^2(1-\lambda)^2} + \frac{16\eta^2\sum_{j=t-q}^{t-1}(j-t)^2\kappa_b^2(j-t)}{\{\sum_{j=t-q}^{t-1}\kappa_b^2(j-t)\}^2\varepsilon^2(1-\lambda)^2} \\
&\overset{(iii)}{\leq} \frac{C_b}{b\varepsilon^2(1-\lambda)^2} + \frac{16(C_b b\eta)^2}{\{\sum_{j=t-q}^{t-1}\kappa_b^2(j-t)\}^2\varepsilon^2(1-\lambda)^2} \quad (15)
\end{aligned}
$$

The $(i)$ holds since $\mathrm{Var}\{\mathbb{I}(p_j > \lambda)\} \leq 1/4$. As for $(ii)$, consider $\mathcal{A}_\lambda = \{w : F(Q(w)) > \lambda\}$. Then by the two-group model we have $\Pr(p_t > \lambda) = \int_{\mathcal{A}_\lambda}\{f_0(w)(1-\pi_t) + f_1(w)\pi_t\}dw$. Hence

$$
|\Pr(p_j > \lambda) - \Pr(p_t > \lambda)| = \int_{\mathcal{A}_\lambda}\left\{f_0(w)|\pi_j - \pi_t| + f_1(w)|\pi_t - \pi_j|\right\}dw \leq 2|j-t|\eta
$$

by our assumptions. And $(iii)$ is directly from the second part of Lemma E.6.

Above all, we finally conclude that for all $\varepsilon > 0$ and sufficiently large $n_1$, we have

$$
\begin{aligned}
\Pr(|\widehat{\pi}_t^\lambda - \pi_t^\lambda| > \varepsilon) &\leq \frac{C_b}{b\varepsilon^2(1-\lambda)^2} + \frac{36 C_b\sqrt{\log n_1}}{b n_1^{1/2}(1-\pi)(1-\lambda)^2\varepsilon^2} + \frac{48\sqrt{\log n_1}}{n_1^{1/2}(1-\pi)(1-\lambda)^2\varepsilon^2} + \frac{16(C_b b\eta)^2}{\varepsilon^2(1-\lambda)^2} \\
&\leq \max\left\{\frac{1}{b}, \frac{\sqrt{\log n_1}}{n_1^{1/2}}, (b\eta)^2\right\}\frac{c}{\varepsilon^2}
\end{aligned}
$$

for some constant $c > 0$.

Thus take $\varepsilon = \max\left\{1/b, \sqrt{\log n_1}/n_1^{1/2}, (b\eta)^2\right\}^{1/2-\gamma}\sqrt{c}$ and we obtain

$$
|\widehat{\pi}^\lambda - \pi^\lambda| \leq c\max\left\{\frac{1}{b}, \frac{\sqrt{\log n_1}}{n_1^{1/2}}, (b\eta)^2\right\}^{1/2-\gamma}
$$

with probability $1 - \max\left\{1/b, \sqrt{\log n_1/n_1}, (b\eta)^2\right\}^{2\gamma}$, where $0 < \gamma < 1/2$. Hence we complete the proof.

