# OpenReview forum: "Real-Time Selection Under General Constraints via Predictive Inference"
_NeurIPS.cc/2024/Conference — NeurIPS 2024 poster_

### Official Review · Reviewer_vWkU · 2024-07-06

**Soundness:** 3
**Presentation:** 2
**Contribution:** 4
**Rating:** 6
**Confidence:** 1

**Summary:**

The paper proposes a method for online sample selection. The authors introduce the concepts of _individual_ and _interactive constraints_, and demonstrate theoretically and empirically that their method satisfies both.

**Strengths:**

The problem seems important and the formulation and approach novel. The authors provide both theoretical guarantees and empirical evidence, in both synthetic and real-world applications, of the effectiveness of their approach. The mathematical formulation seems sound, and the assumptions and theoretical results are clearly stated.

**Weaknesses:**

I am not familiar with the FDR control literature, and had to read parts of the paper (specifically sections 2.2, 2.3 and 4) multiple times to get a gist for the logic of the method and its empirical perfomance. This is reflected in my confidence score. If the paper is accepted, I highly recommend the authors revise the paper to make it easier to follow. A flowchart to illustrate the steps of the algorithm and/or to illustrate the differences between the Oracle and Data-driven selection procedures may be helpful; a toy example could also help. I also suggest including a longer description and/or table of the benchmark methods against which the empirical performance of II-COS was compared.

**Questions:**

- Interactive constraint: Why is this defined only with respect to the correctly selected samples? In the case of, e.g., the diversity of selected samples, is the constraint not intended to represent the diversity of all selections?
- Line 159: Should $R_t = \left( \sum_{i \leq t} \delta_i \right) \lor 1$?

**Limitations:**

The authors discuss limitations of their work.

---

> ### Author Rebuttal · Authors · 2024-08-04
>
> > W1: I am not familiar with the FDR control literature, and had to read parts of the paper (specifically sections 2.2, 2.3 and 4) multiple times to get a gist for the logic of the method and its empirical perfomance....I highly recommend the authors revise the paper to make it easier to follow.
>
>
> **To W1**: Thank you for your thoughtful suggestions, and we apologize for any confusion caused to readers unfamiliar with the related fields.
>
> - First, we indeed have a flowchart of the data-driven II-COS procedure in Figure 4 of Appendix C. Based on your suggestions, we will add an illustration of the oracle version to the flowchart and move it to the main text in a future version. We are also willing to provide a more detailed description of the benchmarks in the main text, include a toy example of our method, and revise the paper to make it easier to follow according to your helpful advice.
> - Second, due to the limit of main text pages , we have provided more details and related works about our proposed method in supplementary materials. Additionally, in Appendix B, we discuss related works on online FDR control and predictive inference, offering an overview of the literature on online multiple testing and related topics, which can be helpful for readers unfamiliar with these fields.
>
> Again we greatly appreciate your valuable suggestions for improving the quality of our paper.
>
> > Q1: Interactive constraint: Why is this defined only with respect to the correctly selected samples? In the case of, e.g., the diversity of selected samples, is the constraint not intended to represent the diversity of all selections?
>
>
> **To Q1**: Nice question! We'd like to make the following explanations:
>
> - First, in our online selection problem, we are only concerned with the characteristics of the correctly selected samples of interest. In fact, the individual constraint (e.g., FSR) needs to be prioritized because we should select the samples that are of interest (i.e., $Y_t\in\mathcal{A}$). If the interactive constraint is applied to all selected samples, rather than just the correctly selected ones,  we may select many uninteresting samples far from the center of  $Y_t\in\mathcal{A}$ to achieve a low similarity. This is not desirable in practice. For example, hiring a obviously unsuitable candidate due to its diversity only.
> - Second, if in other problems and scenarios where constraints need to be applied to all selected samples (rather than just the correctly selected samples), our method can be easily modified to meet this requirement by replacing $1-\widehat{L}$  in Equation (7) with 1 in the decision rule of II-COS.
>
> We hope the above clarification will resolve your concerns. Thank you!
>
>
> > Q2: Line 159: Shluld $R_t=(\sum_{i\leq t} \delta_i)\vee 1$?
>
>
> **To Q2**: Yes, thank you for your meticulous attention to details. We will incorporate this modification into the final version of our paper.

---

> > ### Comment · Reviewer_vWkU · 2024-08-08
> >
> > Thank you for these clarifications.

---

### Official Review · Reviewer_vGU3 · 2024-07-09

**Soundness:** 4
**Presentation:** 4
**Contribution:** 4
**Rating:** 8
**Confidence:** 4

**Summary:**

In this paper, the authors quantify the uncertainty of response predictions using predictive inference; and systematically addressing individual and interactive constraints in a sequential manner.  An online selection rule is developed to ensure the above two types of constraints are under control at pre-specified levels simultaneously.  Simulated and real-data examples  are used to evaluate and illustrate their approach in terms of both online individual and interactive criteria control.

**Strengths:**

This is a nicely, clearly written paper that develops an online selection rule that is simple yet effective.  The simplicity of the online selection rule will enhance the potential for  this rule to be used in real life.   The authors’ claims are well supported both via theory, simulations and application to real data. The paper along with the appendix provides detailed information that allow for replicability.  Very nice!  I particularly appreciate the comparison with the approaches based on conformal p-values.

**Weaknesses:**

See questions below.

**Questions:**

Section 4.1 would be good to tell reader how many replications are used.

**Limitations:**

Comments are needed on the practicality of \hat\mu being a bijection.  How critical is this assumption to the method and the theoretical analysis.  If critical then this is a limitation.

---

> ### Author Rebuttal · Authors · 2024-08-04
>
> > Q: Section 4.1 would be good to tell reader how many replications are used.
>
> **To Q**: We have mentioned '500 replications' in the begining of Section 4 (line 270) when introducing our evaluation measures. To avoid repetition, we did not restate this in Section 4.1.
>
> > L: Comments are needed on the practicality of \hat\mu being a bijection. How critical is this assumption to the method and the theoretical analysis. If critical then this is a limitation.
>
>
> **To L**: Thank you for your nice advice.
>
> - The assumption is considerably mild and widely adopted for the identyification of each $X_t$ in the predictive inference framework. Please refer to [1] for similar assumptions. Per your suggestion, we will add a comment about this assumption in the final version.
> - Besides, even if this assumption fails, we can view the local FDR as a special weight to reflect the correctness of each selection. And our strategy can also provide desirable cost control and diversity exhibition empirically.
>
> Reference:
> [1] Wu, X., Huo, Y., Ren, H., and Zou, C. Optimal subsampling via predictive inference. JASA, 2023.

---

> ### Comment · Reviewer_vGU3 · 2024-08-07
>
> Thanks for your response.  I maintain my rating of 8.

---

### Official Review · Reviewer_MMvB · 2024-07-12

**Soundness:** 3
**Presentation:** 3
**Contribution:** 2
**Rating:** 4
**Confidence:** 4

**Summary:**

The paper studies online sample selection with individual and interaction
constraints simultaneously. Specifically, the goal is to control (variants) of
the false selection rate (FDR) and the expected similarity (ES) under the
empirical bayes framework. Under distributional conditions, the proposed method
controls the target quantities asymptotically. The method is evaluated on synthetic
and real data.

**Strengths:**

1. The paper is well-presented and easy to follow.
2. The problem under consideration is of interest and relevant.

**Weaknesses:**

Please see my comments in the questions section.

**Questions:**

1. **The model.** In the motivating examples such as candidate selection, it appears to me
that we get to observe the ground truth, i.e., $Y_t$, after time step $t$.
This is briefly mentioned in the discussion section, but I think it is reasonable
to use the observed $Y_t$'s to update the estimation.

2. **The choice of interaction constraints.** It is not well-motivated why
the changing ES to (4) is reasonable. As illustrated in the simulation,
although these two quantities seem to coincide as the time step goes
to infinity, they do differ quite a lot with smaller time steps (this could happen
when $m$ is small).

3. What is the principle for choosing K in general? In particular, for the real data example,
why is K taken to be $1\times 10^{-3}$ in the fist example and $6\times 10^{-3}$ in the second?

4. In the numerical experiments, a fairer comparison with offline CP would be with [1], i.e.,
thresholding the p-values with BH-adjusted p-values as opposed to the fixed threshold.

[1] Jin, Ying, and Emmanuel J. Candès. "Selection by prediction with conformal p-values." Journal of Machine Learning Research 24.244 (2023): 1-41.

**Limitations:**

The authors have partically addressed the limitations.

---

> ### Author Rebuttal · Authors · 2024-08-04
>
> > Q1: In motivating examples such as candidate selection, we get to observe the ground truth after time t. ... it is reasonable to use the observed $Y_t$'s to update the estimation.
>
> **To Q1**: Thank you for the constructive suggestion. We'd like to clarify a few points:
>
> 1. **Feedback and Model Updates**: As you mentioned, it's of great interest to explore online sample selection with feedback (past ground truth Y), which is part of our ongoing work. However, using new observations can improve the accuracy of estimated models, but updating these models at every time t can be time-consuming and inefficient for some machine learning algorithms. Additionally, ensuring selection error rate control with feedback requires stability criteria on the model or learning algorithm. This falls into a different regime compared to our current setting. Our method, II-COS, is model-agnostic and compatible with commonly-used learning algorithms. This paper focuses on a general framework without feedback as the first effort in online selection considering general interactive and individual constraints control.
> 2. **Unavailable Responses in Online Settings**: In many cases, the response is not available in online settings. For instance, in online anomaly detection [1], we cannot access the outlier label throughout the procedure. Responses can also be delayed, effectively making ground truth unavailable within the restricted time of the online process. Candidate selection is a special case of this, as identification occurs after resumes have been passed for a period. Hence, our model is suitable for these scenarios.
>
> We hope this clarification addresses your concerns about our model.
>
> Ref: [1] Gang, B., Sun, W., Wang, W. Structure-adaptive sequential testing for online false discovery rate control. JASA, 2023.
>
> > Q2: It is not well-motivated why the changing ES to (4) is reasonable....
>
> **To Q2**: Thanks for your question. We would like to make some explanations.
>
> - First,  the motivation for using mES instead of ES is the technical difficulty of directly dealing with the expectation of a ratio. The mES is a ratio of expectations, which is easier to control and also serves as a reasonable measure of similarity.
> - In fact, mES and ES are inspired by the mFDR (modified false discovery rate)  and FDR in the field of online multiple testing, where mFDR is a ratio of expectations and FDR is the expectations of a ratio. The mFDR is usually employed as a replacement for FDR and is shown asymptotically equivalent to FDR [1]. Similar techniques also can be applied for the asymptotic equivalence for mES and ES.
> - In numerical studies, we see that empirically the two similarity measures yield almost identical patterns when there are sufficient samples. An illustrative example can be found in Figure 6. Besides, in Sec. 4.1, we have shown the online ES values agains time t in Figure 1(right), from which we can see that our II-COS can guarantee valid online ES control even for small values of t.  Therefore, when m is small, our method is also empirically valid for controlling ES.
>
> Ref:  [1] Sun W, Cai T T. , Oracle and adaptive compound decision rules for false discovery rate control, JASA, 2007.
>
> > Q3: What is the principle for choosing K ?
>
> **To Q3**: In Appen C.1 (line 512), we provided implementation guidelines for choosing K. To add more clarity, we'll explain it here.
>
> - For any two i.i.d. observations $X$ and $X'$with corresponding $\theta$ and , the expected $C_2$ of the individuals of interest is given by  ${C_2}=E[g(X,X')\mid\theta=1,\theta'=1]$, which can be estimated by $\widehat{C_2}=\sum\sum_{i<j; i,j\in\mathcal{L}} g({X}_i,{X}_j)/\{|\mathcal{L}|(|\mathcal{L}|-1)\}$, $\mathcal{L}=\\{i:{Y}_i\in\mathcal{A}\\}$.
> - We then set $K=a\widehat{C_2}$, where $a>0$ is user-specific to control interactive constraint level. Our numerical evidence shows that $a\in (0.1,0.5)$ works generally well. And we set a=0.4 in our experiments.
> - For real data examples, since $\widehat{C}_2$ differs between the datasets, we set the same parameter $a=0.4$, leading to different K values for each application. Both choices are reasonable.
>
> > Q4: In experiments, a fairer comparison with offline CP would be with [1], i.e., thresholding the p-values with BH-adjusted p-values as opposed to fixed threshold.
>
> **To Q4**: Thank you for your suggestion. We considered multiple factors when selecting the baselines and chose several that we believe are reasonable. Here are some clarifications:
>
> 1. **Conformal-p-values-based baselines**: We have compared several online FDR control methods using conformal p-values in [1], including LOND, SAFFRON, and ADDIS (see Figures 1-2). Notably, LOND is exactly the online version of the BH procedure, and SAFFRON is the online version of the Storey-BH procedure. More details can be found in [2] and [3]. As mentioned in line 298, for real-data experiments, online multiple testing methods based on conformal p-values resulted in few selected individuals. Therefore, we focused on comparing II-COS with SAST, and omitted results for conformal-p-value-based methods.
> 2. **Online case consideration**: The BH procedure itself cannot be applied in online scenarios as it requires all p-values to be sorted before making a decision, which is not possible when future p-values are unknown. Thus, we compared our method with the fixed threshold method in real-data examples (see Table 1 and Figure 3), as it can be executed in online setting and serves as one of our baselines.
>
> We hope these clarifications address your concerns. We would appreciate it if you could re-evaluate our work.
>
> Refs:
> [1] Jin, Y. and Candes, E. J. Selection by prediction with conformal p-values. JMLR, 2023.
>
> [2] Ramdas, A., Jordan, M. I. et al., Online control of the false discovery rate with decaying memory. NeurIPS, 2017.
>
> [3] Ramdas, A.,  Jordan, M. I. et al., SAFFRON: an adaptive algorithm for online control of the false discovery rate. ICML, 2018.

---

### Official Review · Reviewer_78XN · 2024-07-26

**Soundness:** 3
**Presentation:** 3
**Contribution:** 3
**Rating:** 7
**Confidence:** 4

**Summary:**

This paper introduces a framework to perform online sample selection such that the unseen outcomes are in specific target range while also optimizing for constraints like diversity that are dependent on the input covariates. The additional constraint involving input covariates can help ensure properties like the diversity of candidates when selecting individuals for interviews while also guaranteeing that most of the interviewed individuals accept the offer. The paper proposes a data-driven procedure to select the subset of candidates in an online fashion by implementing the proposed algorithm. Under reasonably weak assumptions, the paper provides theoretical guarantees on satisfying both the above constraints in online sample selection. The experiments confirm that this framework ensures low false selection rates (i.e. unseen outcomes are in a specific target range) while optimizing for the additional covariate-dependent constraints like diversity on synthetic and real data.

**Strengths:**

The paper proposes an intuitive way to incorporate covariate-dependent constraints like the diversity of candidates when performing online sample selection to optimize for metrics like false selection rates. This paper solves an important problem in online sample selection and demonstrates that the proposed method improves the covariate-dependent objective while maintaining comparable performance on false selection rates.

**Weaknesses:**

It would be interesting to understand the gaps between an ideal diversity profile and the profile obtained by the proposed method in Fig 3. Analysing the gap w.r.t changing g(X_i, X_j) function choice could be helpful. Would it be helpful to increase the weight of the g(X_i, X_j) term to reduce this gap and understand its implications on the satisfaction of individual constraints?

It is evident that the SAST baseline outperforms the proposed method in terms of FSR sometimes, which is understandable given there; 's a tradeoff with the interactive constraints (Table 2b, Fig 1).  It would be helpful to learn if we can reduce the gap between SAST and the proposed method by balancing the tradeoff (perhaps using a tunable hyperparameter that balances the two constraints?).

**Questions:**

See weaknesses

**Limitations:**

Yes, limitations and broader impacts are discussed in the last section of paper.

---

> ### Author Rebuttal · Authors · 2024-08-04
>
> > W1: It would be interesting to understand the gaps between an ideal diversity profile and the profile obtained by the proposed method in Fig 3. Analysing the gap w.r.t changing g(X_i, X_j) function choice could be helpful. Would it be helpful to increase the weight of the g(X_i, X_j) term to reduce this gap and understand its implications on the satisfaction of individual constraints?
>
>
> **To W1**: Thank you for your thoughtful suggesstions! We'd like to make some explanations as follows:
>
> - First, your suggestion is very reasonable, especially regarding our focus on specific groups such as Bachelor's degrees. As the categorical variable is transformed into one-hot coding to numerically compute the diversity, we can assign greater weights to these specific groups (one-hot variables) of interest. This adjustment allows us to enhance their contribution to achieving diversity.
> - Second, defining an ideal diversity profile precisely is challenging. Since changing g affects how we define our target diversity criterion. From our perspective, the ideal diversity profile can be defined as achieving maximal diversity in the offline setting while selecting the same number of units and controlling the FSR. A related work is [1].  In this aspect, the gap primarily arises due to differences in the online procedure and cannot be easily analyzed for its relationship with g.
> - Third, we conducted an experiment on candidate data to demonstrate the diversity performance using our II-COS method, the offline oracle, and your innovative weighting approach, respectively. Here we assign more weight on Bachelor degree (five times than other education variable). We fix $\alpha=0.2, K=0.001$. The education status composition of the correctly selected samples and FSR for three methods are shown in the table below. We can observe that this weighting scheme effectively increased the proportion of Bachelor's degrees, aligning closer to the distribution observed in the offline oracle. However, it also resulted in a decrease proportion in Master's degrees, which is less desirable. Therefore, determining an optimal weighting scheme to achieve the ideal diversity profile remains an interesting question for future investigation.
> - As for the implications on individual constraints, we find no significant influence empirically when adjusting the weight of g, since the FSR control of our strategy is quite tight.
>
>  Hope these explainations are acceptable for you!
>
> |  | FSR | No Qual | High School |  Matriculation   |  Bachelor  | Master |
> | --- | --- | --- | --- | --- | --- | --- |
> | Oracle |  0.19 | 0.03 | 0.34 | 0.26 | 0.27 | 0.08 |
> | II-COS | 0.19 | 0.03 | 0.44 | 0.26 | 0.21 | 0.07 |
> | More weight on Bachelor | 0.19 | 0.02 | 0.39 | 0.29 | 0.28 | 0.02 |
>
>
> Reference:
> [1] Wu, X., Huo, Y., Ren, H., and Zou, C. Optimal subsampling via predictive inference. JASA, 2023.
>
>
> > W2: It is evident that the SAST baseline outperforms the proposed method in terms of FSR sometimes, which is understandable given there 's a tradeoff with the interactive constraints (Table 2b, Fig 1). It would be helpful to learn if we can reduce the gap between SAST and the proposed method by balancing the tradeoff (perhaps using a tunable hyperparameter that balances the two constraints?).
>
>
> **To W2**:  Thank you for your questions! Your observation is very insightful, and in fact, our proposed method already can achieve the trade-off you mentioned.
>
> - First, our method itself provides a sufficiently flexible framework that allows for balancing this trade-off by adjusting the parameters $\alpha$ and K as needed. Typically, one could choose $K=+\infty$ for the case the interactive constraint is out of work and only individual constraint control is considered, and meanwhile one can set $\alpha=1$ with which the interactive constraint is the only concern. The results in Table 2 (Appendix E.2) evaluated this conclusion.
> - Second, as mentioned in Section 2.2 (lines 169-171) of the paper, SAST can be regarded as a special case of our proposed method. When the individual constraint is set to FSR and K is set to $+\infty$ , our method reduces to the same manner as the controlling step of the SAST.
> - Therefore, additional tuning parameters are not specifically required to achieve this.
>
> We hope that this interpretation can ease your doubts.

---

> > ### Comment · Reviewer_78XN · 2024-08-12
> >
> > Thank you for the detailed response and additional work on experiments.
> >
> > I am increasing my rating to accept (7).

---

### Decision · Program_Chairs · 2024-09-25

**Decision:**

Accept (poster)

**Comment:**

This paper explores an online sample selection problem with both individual constraints (e.g., false selection rate) and interactive constraints (e.g., diversity). The authors propose a simple yet effective algorithm, supported by strong theoretical analysis and numerical experiments. The reviews were generally positive, and the rebuttals effectively addressed the reviewers' questions. Therefore, I recommend acceptance.

For the camera-ready version, I encourage the authors to incorporate their discussions with the reviewers and enhance the presentation as suggested by reviewer vWkU.